# How Far Can LLM Agents Reason with Tables?
# Benchmarking Multi-Turn Agentic Table Question Answering in the Wild

**Jingwang Huang** [* 1]  **Jie Zhang** [* 2]  **Haoyang Zeng** [* 1]  **Changzai Pan** [2]  **Xianjie Wu** [3]  **Guanting Dong** [4]
**Jiaheng Liu** [5]  **Wei Zhang** [3]  **Mingyu Zheng** [6]  **Chunxiao Liu** [1]  **Kaiwen Wei** [1]  **Jiang Zhong** [1]  **Jian Yang** [3]

## Abstract

Recent advances in large language models (LLMs) have substantially expanded the scope of Table Question Answering (TableQA). However, existing benchmarks primarily treat TableQA as a passive, single-turn natural language understanding task, lacking the capacity to evaluate autonomous reasoning and tool-call trajectories in realistic, multi-turn scenarios. To bridge this gap, we introduce TableAgent-Bench, a large-scale bilingual benchmark that reformulates TableQA as proactive, agentic interactions over structurally complex, multi-table environments. With a topology-aware construction strategy, TableAgent-Bench captures dynamic intent evolution through 1,310 multi-turn dialogues grounded in 2,275 industrial tables. Furthermore, we propose the Table-centric Agent Evaluation Framework (TAEF) to assess agent interactions with complex table structures. Specifically, TAEF integrates a specialized agent toolset and 4 metric categories to systematically diagnose intermediate failure modes, assessing performance across table localization, tool-invocation rationality, and trajectory-level pass rate. Extensive experiments with 25 state-of-the-art LLM agents reveal a substantial capability gap, with even the strongest model Gemini-3-Pro-Preview achieving only 53.4% information coverage. We expect TableAgent-Bench to serve as a rigorous testbed

for developing and evaluating agents capable of robust table-centric reasoning.

## 1. Introduction

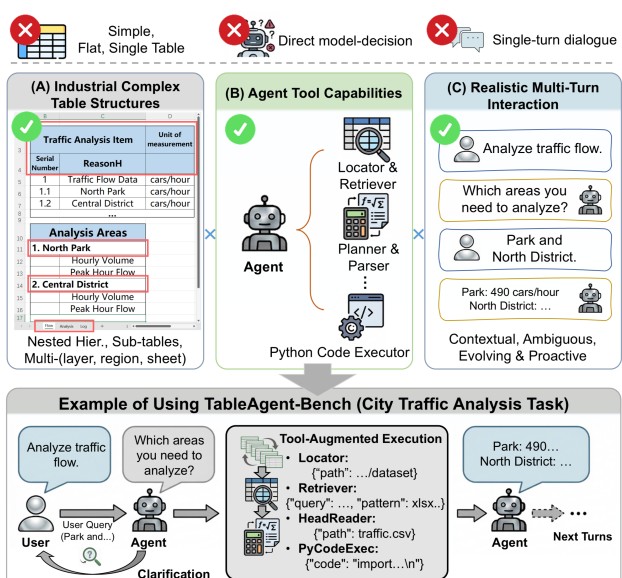

*Figure 1.* TableAgent-Bench considers key agentic challenges in real-world TableQA, including complex table structures, seamless tool integration, and sophisticated multi-turn dialogue interactions.

Tabular data has long served as a fundamental medium for structured information representation. Propelled by the reasoning and tool-use prowess of Large Language Models (LLMs) (Shi et al., 2025; Zhang et al., 2025b), recent breakthroughs in table understanding (Sui et al., 2024; Zhou et al., 2025) and reasoning (Lu et al., 2024; Zhang et al., 2024) have significantly broadened the scope of Table Question Answering (TableQA). This evolution facilitates sophisticated decision-making in real-world domains like business intelligence (BI) and enterprise systems.

To evaluate existing TableQA methods, various benchmarks have been developed. Early datasets like WikiTableQuestions (Pasupat & Liang, 2015) and TabFact (Chen et al., 2019) primarily address single-turn, retrieval-based reasoning. Although subsequent works have introduced domain-

---

[*]Equal contribution [1]College of Computer Science, Chongqing University, Chongqing, China [2]China Telecom Artificial Intelligence Technology (Beijing) Co., Ltd [3]School of Computer Science and Engineering, Beihang University, Beijing, China [4]Gaoling School of Artificial Intelligence, Renmin University of China, Beijing, China [5]College of Intelligent Science and Technology, Nanjing University, Suzhou, China [6]Institute of Information Engineering, Chinese Academy of Sciences, Beijing, China. Correspondence to: Kaiwen Wei <weikaiwen@cqu.edu.cn>, Jiang Zhong <zhongjiang@cqu.edu.cn>, Jian Yang <jiaya@buaa.edu.cn>.

*Proceedings of the 43rd International Conference on Machine Learning*, Seoul, South Korea. PMLR 306, 2026. Copyright 2026 by the author(s).

*Table 1.* Comparison between TableAgent-Bench and existing representative Table benchmarks. Given the scarcity of industrial data in public datasets, we restrict our comparison to benchmarks that incorporate industrial tables.

| Benchmark | Task Type | Multiple Tables | Complex Structure Tables | Multi-turn Dialogue | Language | Tables Number | Total Multi-turn Dialogue Examples | Domains |
|---|---|---|---|---|---|---|---|---|
| TAT-QA(Zhu et al., 2021a) | TableQA | × | × | × | EN | 20000 | - | 1 |
| AIT-QA(Katsis et al., 2022) | TableQA | × | √ | × | EN | 116 | - | 1 |
| HiTab(Cheng et al., 2022) | TableQA | × | √ | × | EN | 3597 | - | 29 |
| TableBench(Wu et al., 2024) | TableQA | × | × | × | EN | 586 | - | 18 |
| DataBench(Grijalba et al., 2024) | TableQA | × | × | × | EN | 165 | - | 8 |
| SciTableQA(Ajayi et al., 2025) | TableQA | × | √ | × | EN | 320 | - | 5 |
| MiMoTable(Li et al., 2024) | TableQA | √ | √ | × | ZH,EN | 428 | - | 7 |
| GRI-QA(Contalbo et al., 2025) | TableQA | √ | √ | × | EN | 204 | - | 7 |
| MMTU(Xing et al., 2025) | TableQA | √ | √ | × | EN | 61,763 | - | – |
| RealHiTBench(Wu et al., 2025b) | TableQA | √ | √ | × | EN | 708 | - | 24 |
| ToTTo(Parikh et al., 2020) | Table2Text | × | × | × | EN | 83,141 | - | 44 |
| T2R-bench(Zhang et al., 2025c) | Table2Report | √ | √ | × | ZH,EN | 457 | - | 19 |
| TableAgent-Bench (Ours) | TableQA (Multi-turn) | √ | √ | √ | ZH,EN | 1549(ZH), 726(EN) | 1310 | 28 |

specific (Zhu et al., 2021b) and structurally complex tables (Wu et al., 2024), they largely treat TableQA as a passive natural language understanding task. In contrast, general-purpose agentic benchmarks (Farn & Shin, 2023; Wang et al., 2024) emphasize proactive reasoning trajectories, requiring agents to autonomously plan, tool call, and adapt to multi-turn dialogue with feedback. Despite this progress, the transition from general-domain agent to table-specific agentic reasoning remains underexplored.

This gap prompts a fundamental question: *How far can LLM-based agents autonomously navigate and reason with tables in the wild?* As illustrated in Fig. 1, bridging the gap between traditional TableQA and agentic workflows involves two primary challenges: (1) *Benchmarking challenge*. Real-world table interactions are inherently agentic and driven by multi-turn intent evolution. However, existing benchmarks remain confined to decontextualized, single-turn queries over academic tables, leaving critical agentic requirements including reasoning over interdependent sub-questions and complex table relationships largely underexplored. (2) *Infrastructure and Evaluation Challenge*. Industrial-grade assessment necessitates a unified infrastructure to both facilitate and evaluate agentic behaviors beyond terminal accuracy. Current frameworks lack the specialized toolsets and sandboxed environments required for interaction with complex table understanding in multi-turn scenarios. Consequently, current protocols lack the capacity to diagnose intermediate failure modes such as table localization and tool-invocation rationality. This absence makes it hard to discern whether agent trajectories are logically sound or merely fortunate within expansive, multi-table environments.

To address these challenges, we introduce **TableAgent-Bench**, a large-scale bilingual benchmark for evaluating LLM-based agents on multi-turn TableQA in industrial settings. Specifically, we employ a topology-aware construction strategy utilizing 6 question structures and 10 templates to capture complex inter-table dependencies, resulting in 1,310 dialogues grounded in 2,275 tables. Furthermore, we propose the Table-centric Agent Evaluation Framework

(**TAEF**). TAEF facilitates agent-centric assessment through a specialized toolset modeling human-like workflows and incorporates 4 metric categories to systematically diagnose performance in sub-question quality, trajectory-level quality, table localization capability, and tool invocation behavior. Experiment results reveal a substantial capability gap, as even the strongest model Gemini-3-Pro-Preview achieves only 53.4% information coverage and 5% Avg@3. This underscores the complexity of autonomous reasoning in realistic tabular environments. In summary, the contributions of this paper are as follows:

(1) We introduce TableAgent-Bench, a large-scale bilingual benchmark featuring a topology-aware dialogue construction strategy. It evaluates LLM agents on multi-turn TableQA within complex industrial environments, encompassing 1,310 dialogues grounded in 2,275 tables.

(2) We propose the TAEF, an automated evaluation framework that integrates an agent toolset and 4 carefully designed metric categories to systematically assess agents' performance in multi-turn, table-centric reasoning.

(3) We conduct extensive experiments with 25 state-of-the-art LLM-based agents. Results reveal a substantial capability gap, with the leading model achieving only 5% Avg@3, thereby underscoring the challenges of agentic table reasoning in realistic scenarios. The data and code are available at https://201983290498.github.io/TableAgentBench.

## 2. Related Work

**TableQA Benchmarks.** TableQA has evolved from foundational datasets such as WikiTableQuestions (WTQ) (Pasupat & Liang, 2015) and TabFact (Chen et al., 2019), which rely primarily on Wikipedia-sourced tables, to benchmarks targeting specialized domains, including aviation (AIT-QA (Katsis et al., 2022)) and finance (TAT-QA (Zhu et al., 2021b), FinQA (Chen et al., 2021)). More recent efforts have begun to address structural complexity; for example, TableBench (Wu et al., 2024) introduces real-world challenges, while MiMoTable (Li et al., 2024) explores multi-scale spreadsheet structures. Further expanding the scope

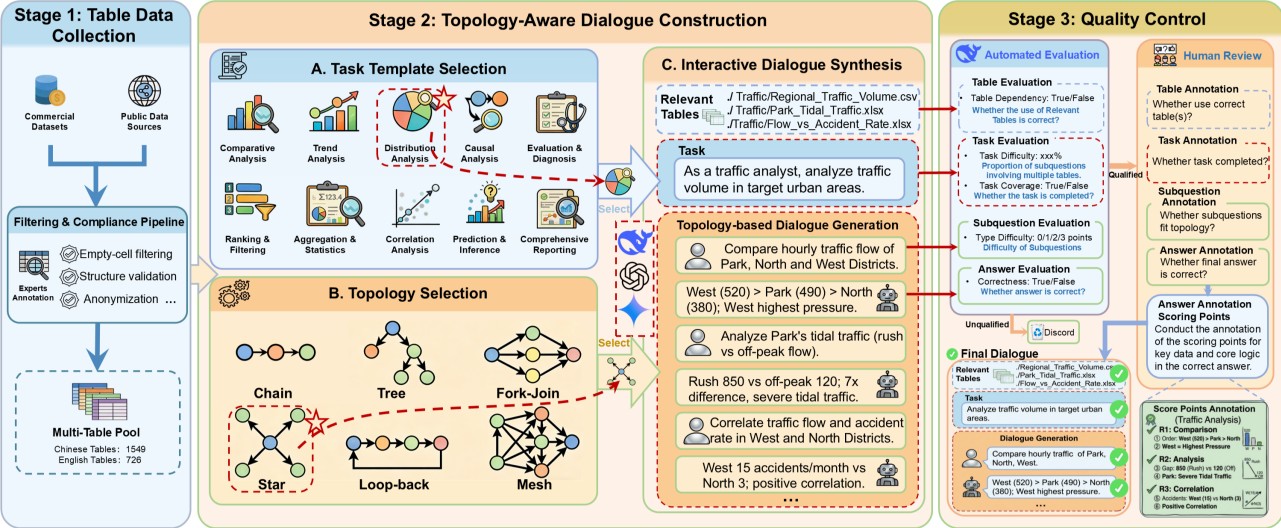

*Figure 2.* An overview of the construction pipeline for TableAgent-Bench, consisting of 3 stages: (1) Table Data Collection, which gathers and filters complex multi-table data from public and commercial sources; (2) Topology-Aware Dialogue Construction, which combines task templates and topology structures to generate multi-turn, table-centric dialogues; and (3) Quality Control, which applies automated evaluation and human review to ensure correctness, coherence, and alignment with task objectives.

of TableQA, TQA-Bench (Qiu et al., 2024) addresses multi-table question answering, focusing on scalable contexts and symbolic extension. Several benchmarks extend TableQA to multi-table settings, including RealHiTBench (Wu et al., 2025a), which focuses on join operations, and MMTU (Xing et al., 2025), which primarily features web-sourced tables. While ReasonTabQA (Pan et al., 2026) targets complex scientific and industrial scenarios, it remains limited to static, single-turn tasks, failing to evaluate agentic behaviors in dynamic, multi-turn dialogues. To fill this gap, we introduce TableAgent-Bench, designed specifically to assess LLM agents within realistic, multi-turn dialogue settings.

**Agent Benchmarks.** Recent benchmarks evaluate LLM agents via multi-turn tool call and reasoning trajectories. General-purpose frameworks, including ToolTalk (Farn & Shin, 2023), MINT (Wang et al., 2024), DialogTool (Wang et al., 2025a), $\tau^2$-Bench (Barres et al., 2025), and VitaBench (He et al., 2025), prioritize conversational tool use and task completion. In data analytics, DAComp (Lei et al., 2025) and DataSciBench (Zhang et al., 2025a) target enterprise workflows and code-based analysis. Parallelly, spreadsheet agents have evolved from single-LLM systems (Li et al., 2023; Wang et al., 2025b) to multi-agent collaborative frameworks (Chen et al., 2025; Tian et al., 2025; Yu & Chen, 2025; Zhu et al., 2025). However, these systems often lack autonomous file-system retrieval and complex cross-table analysis, failing to sustain the long-horizon tool trajectories essential for industrial tasks. Moreover, existing evaluations often treat the intermediate reasoning within trajectories as a black box by prioritizing terminal accuracy over fine-grained execution failures. This opacity is

exacerbated by a reliance on LLM-as-a-judge paradigms or predefined trajectories, which compromise objectivity in complex environments. While specialized efforts (Deshpande et al., 2025; Lu et al., 2025; Schmidgall et al., 2024; Tu et al., 2024) introduced diverse protocols for agent capability assessment, a comprehensive assessment of agents in industrial, multi-turn, and long-horizon tabular environments remains an open challenge.

## 3. TableAgent-Bench Construction

As illustrated in Figure 2, the construction process of TableAgent-Bench comprises 3 main stages: table data collection, dialogue construction, and quality control.[1]

### 3.1. Table Data Collection

During table collection, we emphasize source diversity and structural complexity to support realistic analytical scenarios across domains. Tables are collected from public online sources and commercially acquired datasets, covering diverse industries and structural forms, including multi-table collections and complex headers. Rather than randomly combining unrelated tables, we organize the collected tables into scenario-level folders that preserve the natural physical or semantic boundaries of the original data sources. These folders typically correspond to multi-sheet enterprise reports, tables released under the same public statistical program, or thematic industrial reports. As a result, tables

---

[1]More details on the construction process, and the synthesis or evaluation methodology are provided in Appendix C.

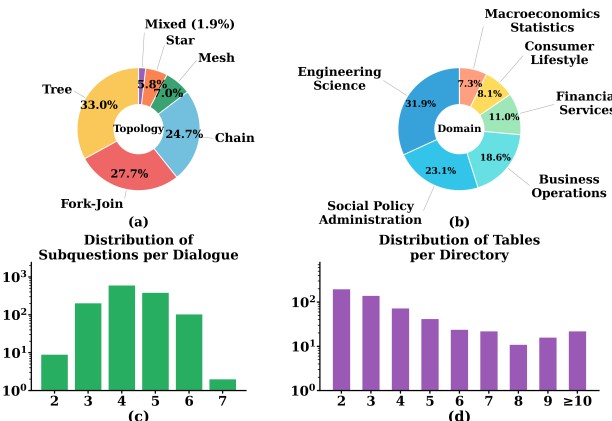

*Figure 3.* Statistics of the TableAgent-Bench. (a) Topology distribution. (b) Domain distribution. (c) Distribution of the average number of sub-questions in each multi-turn dialogue. (d) Distribution of tables in each directory.

*Table 2.* Key Statistics of TableAgent-Bench.

| PROPERTY | VALUE |
| --- | --- |
| NUMBER OF CHINESE TABLES | 1,549 |
| NUMBER OF ENGLISH TABLES | 726 |
| AVERAGE TABLES PER FOLDER | 4.15 |
| TOTAL MULTI-TURN DIALOGUE EXAMPLES | 1,310 |
| TOTAL SUB-QUESTIONS | 5,617 |
| AVERAGE NUMBER OF DIALOGUE TURNS | 4.29 |

within the same folder provide a weak but meaningful structural prior for table retrieval and cross-table analysis, while the exact task-relevant table set is further verified during quality control.

All tables undergo manual pre-screening to ensure coverage and relevance, followed by automated filtering to remove sparse, weakly structured, or noisy tables. Sensitive content is detected and anonymized in compliance with data licenses and usage agreements. In total, we collect 1,549 Chinese and 726 English tables across 6 domains and 28 subdomains, forming the foundation of TableAgent-Bench.

### 3.2. Topology-Aware Dialogue Construction

Real-world user intent is typically expressed through complex, goal-driven multi-turn interactions (Budzianowski et al., 2020; Li et al., 2025). To model this behavior, we introduce *Task* as a global object that governs and constrains dialogue evolution, with task type sampled from 10 predefined templates to ensure topical diversity. Inspired by human cognitive patterns in multi-table analysis, we further abstract 6 inter-question topology structures to capture cross-turn dependencies, thereby distinguishing multi-turn dialogues from single-turn QA and enabling scalable dialogue synthesis with LLMs.

**Task Template Selection.** To enhance task and dialogue diversity, we derive ten distinct analysis task templates (e.g., Comparative Analysis, Trend Analysis) from real-world multi-table analytical contexts and design specialized templates for each scenario. Each task is instantiated by sampling one template from this collection, ensuring rich variation in the resulting conversation.

**Topology Selection.** Upon task template, we define six typical topology structures (Chain, Tree, Fork-Join, Star,

Loop-back, and Mesh) to support diverse multi-turn conversations. These topologies were selected based on common patterns observed in real-world multi-turn dialogue and planning tasks (Wang et al., 2023). Each topology is modeled as a directed dependency graph, where nodes represent sub-questions and edges encode their dependencies. This structure constrains cross-turn information flow and guides the dialogue progression toward the task objective in multi-table settings.

Topology selection is task-dependent rather than orthogonal; different analytical tasks favor specific structural patterns. For example, *Distribution Analysis* task, which require cross-table data comparison, align with the *Star* topology, where a central theme branches into multiple independent queries. These task–topology correspondences are systematically characterized in Appendix C.6, and compatible pairs are randomly sampled for each table set to form the structural basis for dialogue synthesis.

**Interactive Dialogue Synthesis.** Given a task–topology pair and source tables, we employ LLMs (DeepSeek-V3.2, GPT-5.2, and Gemini-3-Pro) to generate 2–6-turn dialogue skeletons. For identical source table sets, multiple dialogue candidates are produced by varying task types, topologies, and LLM generators to enhance diversity. This redundancy creates an ample initial pool for manual curation while enhancing the richness and variety of multi-turn interactions. To enhance overall fluency and naturalness, the dialogues are further augmented via coreference resolution, ellipsis restoration, and paraphrasing, exhibiting diversity in both structure and content, closely mirroring real human user interactions. The final output consists of source tables path, task and complete multi-turn QA sequences, enabling precise evaluation of model reasoning and tool call in complex tabular contexts.

### 3.3. Quality Control

To ensure the correctness and reliability of generated dialogues, we adopt a two-stage quality control pipeline comprising *automated evaluation* and *human review*.

In the automated stage, DeepSeek V3.2 conducts preliminary screening by assessing table source dependency consistency, task completeness, question validity, and answer correctness and plausibility, while filtering out overly sim-

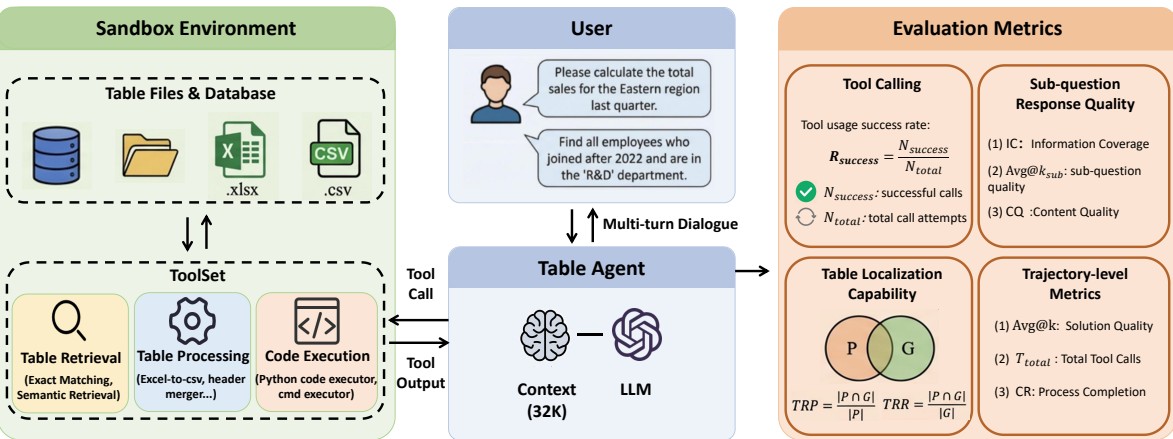

*Figure 4.* Overview of the TAEF framework. TAEF comprises three core components: a *User* issuing multi-turn sub-questions; a *Table Agent* that addresses these through table localization, structure interpretation, and data analysis via tool invocation; and a *Sandbox Environment* hosting both Table Files Database and the ToolSet. The *Evaluation Metrics* assess agent performance across four key dimensions: tool calling, sub-question response quality, table localization capability, and trajectory-level metrics.

plistic instances.

In the human review stage, experts manually audit the filtered dialogues to correct residual errors. The annotation process focuses on 4 key dimensions: (1) accuracy of relevant table paths and the dependencies between questions and tables; (2) consistency between tasks and multi-turn dialogues, ensuring the dialogue effectively achieves the task objectives; (3) clarity of sub-questions and their adherence to the specified topology; and (4) factual correctness of sub-question answers, ensuring they are correct answers. For each sub-question, experts construct reference answers based on the preliminary automated evaluations, with comparative analysis revealing high consistency between these human-reviewed answers and the automated results (Cohen's Kappa = 0.873). To handle the open-ended nature of multi-turn responses, reviewers establish scoring points grounded in these references, focusing on dimensions such as numerical accuracy and trend interpretation. These points provide structured criteria for the evaluation metrics detailed in Section 4.2.

### 3.4. Dataset Statistics

**Statistics.** TableAgent-Bench consists of 1,310 high-quality dialogues based on 2,275 unique tables. Each dialogue includes source table paths, task, complete multi-turn QA sequences, and expert-annotated critical scoring points, enabling systematic evaluation of model capabilities in multi-step planning, cross-table reasoning, and tool use. Key statistics are provided in Table 2 and Figure 3.

**Domain Coverage.** TableAgent-Bench spans 6 primary domains and 28 subdomains, covering diverse real-world contexts such as engineering science, financial services, and consumer lifestyle, thereby mitigating domain-specific bias.

**Table and Dialogue Scale.** Each instance involves multiple tables (4.15 on average) and multi-turn dialogues averaging 4.29 turns, yielding a substantially higher proportion of multi-table scenarios than traditional single-table QA benchmarks and better reflecting real-world analytical workflows.

**Topology Distribution.** TableAgent-Bench covers all 6 topology types: Chain (24.58%), Tree (32.88%), Fork-Join (27.63%), Star/Loop-back/Mesh (collectively 13.05%), and Hybrid (1.86%). This distribution reflects a range of real-world interaction patterns, from linear questioning to more complex structures such as branching, parallel comparison, and retrospective correction.

## 4. TAEF for TableQA

As illustrated in Figure 4, we propose the Table-centric Agent Evaluation Framework (TAEF), an automated framework simulating and evaluating real-world data analysis where agents resolve complex queries through multi-round interactions over heterogeneous table collections. It comprises three core components: *a User* sequentially issuing sub-questions, *a Table Agent* responding sub-questions by invoking sandbox-hosted tools, and *a Sandbox Environment* as an isolated workspace hosting tables and tool interfaces.

At the beginning of each task, the agent receives only the current user query, dialogue history, and the root directory of the table database, rather than pre-filtered relevant tables or table summaries. It must autonomously inspect the file system, identify candidate tables, read table contents, and perform analysis through tool calls. Scenario-level folders provide only a weak structural prior for retrieval, while the

ground-truth relevant tables used for localization evaluation are determined by human verification. Upon completion of each dialogue, the resulting multi-turn, multi-step trajectory is recorded. This trajectory provides a comprehensive log of the interaction process, including sub-questions, tool invocations, tool feedback, and agent thinking steps, and forms the basis for subsequent evaluation of agent performance and dialogue strategies.

### 4.1. Table-Centric ToolSet Design

When working with multiple tables, human analysts typically follow a 3-stage workflow: *locating relevant tables, analyzing and understanding their contents, and synthesizing answers*. To equip agents with analogous capabilities, we propose our toolset into 3 functional categories (see appendix D.1 for details).

**Table Retrieval Tools.** These combine exact matching (**EM**) and semantic retrieval (**SR**) capabilities. Exact matching based on regular expressions enables rapid string-level localization within tables, while semantic retrieval powered by the Qwen3-Embedding-0.6B (Zhang et al., 2025d) model supports vectorized recall of tables, rows, columns, or cells.

**Table Processing Tools.** The Table Processing (**TP**) tools provide structured multi-table data via Excel-to-CSV conversion, multi-level header merging, and table decomposition. The latter transforms non-standard layouts, such as interleaved blocks and nested hierarchies, into well-structured independent sub-tables.

**Code Execution Tools.** Code Execution (**CE**) tools provide foundational capabilities, including a Python code executor and cmd executor.

### 4.2. Evaluation Metrics

To evaluate the agent's performance, we assess its interaction trajectories along 4 key dimensions: tool calling, sub-question response quality, table localization capability and trajectory-level metrics.

**Tool Calling.** Tool call success rate ($R_{\text{success}}$) is used to evaluate agent tool-call capability across all tasks in the proposed TableAgent-Bench, defined as $R_{\text{success}} = N_{\text{success}}/N_{\text{total}}$, where $N_{\text{success}}$ denotes number of tool calls successfully executed without runtime errors across all tasks, and $N_{\text{total}}$ is total tool-call attempt number across all tasks.

**Sub-question Response Quality.** The quality of responses in multi-turn sub-question answering is evaluated using two metrics: Information Coverage (**IC**) and Content Quality (**CQ**). IC measures the alignment between the model's sub-question responses and human-annotated score points across dimensions such as numerical values, conclusions, and trends. CQ assesses whether the response contains low-

*Table 3.* Overview of the ToolSet.

| Tool Name | Functionality |
|---|---|
| **Table Retrieval Tools** | |
| grep search | Keyword-based search via regular expressions to locate table files matching user queries. |
| table selector | Semantic retrieval of relevant tables using vector embeddings of table metadata. |
| semantic row retriever | Retrieves specific rows via semantic matching between the query and row contents. |
| semantic column retriever | Retrieves columns via semantic matching between the query and column definitions. |
| **Table Processing Tools** | |
| xlsx-to-csv converter | Converts Excel (`.xlsx`/`.xls`) files to CSV, handling multi-sheet files and unmerging cells. |
| table reader | Reads and previews a specified number of rows from a table file. |
| header merger | Flattens complex hierarchical headers (merged cells) into a single row with structure retained. |
| table decomposer | Decomposes non-standard table layouts (e.g. interleaved blocks, nested hierarchies) into independent, well-structured sub-tables via LLM. |
| **Code Execution Tools** | |
| cmd executor | Executes command-line instructions for file system operations. |
| python executor | Executes Python code (e.g., `pandas`) in a sandbox for complex data analysis and computation. |

quality content, including answer-internal consistency or task-irrelevant redundancy (see Appendix D.3). Given that our multi-turn interactive tasks exhibit a mix of deterministic and open-ended characteristics (e.g., responses may contain concise numerical results as well as extended analytical conclusions), traditional n-gram overlap metrics such as ROUGE-L (Lin, 2004) and BLEU (Papineni et al., 2002) are unsuitable. Both metrics are adjudicated by a panel of three LLM-as-Judges (GPT-5, Qwen-Max, and DeepSeek-V3.2) via majority voting (Zheng et al., 2023).

**Table Localization Capability.** To validate the accuracy of relevant table retrieval in multi-table scenarios, we propose two evaluation metrics: Table Relevance Recall (**TRR**) and Table Relevance Precision (**TRP**). The TRP and TRR are computed as follows:

$$\text{TRP} = \frac{|P \cap G|}{|P|}, \quad \text{TRR} = \frac{|P \cap G|}{|G|},$$

where $G$ denotes the ground truth set of tables required for the multi-table task, and $P$ denotes the set of tables retrieved by the model during its reasoning process.

**Trajectory-level Metrics.** We evaluate agent trajectories along 3 complementary dimensions. (1) Solution quality. We adopt **Avg@k** (He et al., 2025) to assess solution quality, which measures the average fraction of perfectly correct solutions over k attempts per task. We also calculate Avg@k for the sub-question level evaluation, denoted as **Avg@k$_{\text{sub}}$**. IC and Avg@k are complementary: IC provides partial

*Table 4.* Overall performance comparison on TableAgent-Bench between thinking and non-thinking modes. Among all metrics, **IC** and **TRR** are relatively important metrics. IC quantifies the average coverage of score points across all queries, while TRR measures the recall of relevant tables. Higher TRR ensures more comprehensive data input for LLM-based analysis and answer generation.

| Models | Sub-question Response Quality | | | Trajectory Level | | | Table Location | | Tool Call |
|---|---|---|---|---|---|---|---|---|---|
| | IC (%)↑ | Avg@3$_{sub}$ (%)↑ | CQ↑ | Avg@3 (%)↑ | CR (%)↑ | $T_{total}$ ↓ | TRR (%)↑ | TRP (%)↑ | $R_{success}$ (%)↑ |
| **Thinking Models** | | | | | | | | | |
| Gemini-3-Pro-Preview | **53.40** | **28.25** | **6.79** | **5.00** | **100.00** | 16.37 | **85.30** | **89.10** | **94.57** |
| Claude-Haiku-4-5 | 51.10 | 24.38 | 5.57 | 0.00 | 97.50 | 36.62 | 76.50 | 74.50 | 94.24 |
| GPT-5 | 50.20 | 24.29 | 6.23 | 5.00 | **100.00** | 16.23 | 83.80 | 82.10 | 90.77 |
| GLM-4.7 | 47.40 | 26.11 | 5.75 | 4.68 | 98.12 | 28.68 | 81.80 | 83.50 | 83.60 |
| DeepSeek-V3.2-Thinking | 46.80 | 27.12 | 6.42 | 0.00 | **100.00** | 35.25 | 74.80 | 74.60 | 89.86 |
| Mimo-V2-Flash | 45.60 | 25.48 | 5.96 | 1.19 | 94.65 | 32.87 | 80.10 | 82.50 | 82.29 |
| Kimi-K2-Thinking | 45.50 | 25.42 | 5.91 | 5.00 | **100.00** | 22.15 | 82.90 | 79.80 | 88.86 |
| GLM-4.6 | 43.83 | 22.79 | 5.93 | 4.17 | 99.17 | 27.82 | 79.20 | 80.80 | 83.35 |
| GLM-4.7-Flash | 39.30 | 25.74 | 5.04 | 0.00 | **100.00** | 27.32 | 73.40 | 71.40 | 90.46 |
| Qwen3-235B-A22B-Thinking | 36.50 | 17.80 | 6.12 | 1.25 | **100.00** | 18.05 | 68.85 | 75.90 | 82.56 |
| Tongyi-DeepResearch-30B-A3B | 15.50 | 13.03 | 2.75 | 0.00 | 45.66 | 61.13 | 36.50 | 20.30 | 88.20 |
| Qwen3-32B | 5.30 | 1.79 | 1.47 | 0.00 | 79.62 | 35.69 | 54.50 | 57.30 | 67.20 |
| Qwen3-14B | 5.15 | 1.61 | 1.71 | 0.00 | 83.92 | 39.54 | 52.90 | 60.70 | 22.17 |
| MiroThinker-v1.5-30B | 3.85 | 1.79 | 0.99 | 0.00 | 40.00 | 65.29 | 26.65 | 29.05 | 87.60 |
| Qwen3-8B | 2.75 | 1.31 | 1.53 | 0.00 | 43.18 | 63.14 | 48.80 | 55.50 | 38.38 |
| **Non-Thinking Models** | | | | | | | | | |
| Claude-Haiku-4-5 | **41.00** | **22.60** | 5.31 | 0.00 | 80.00 | 40.20 | 72.20 | 73.00 | 43.72 |
| Qwen3-Coder-Plus | 39.70 | 18.00 | 5.56 | **2.84** | 98.30 | 31.32 | **76.90** | **79.80** | 86.85 |
| Kimi-K2-0905-Preview | 35.70 | 18.65 | 5.26 | 1.35 | 97.50 | 29.57 | 69.55 | 72.50 | 64.82 |
| Mimo-V2-Flash | 34.90 | 17.50 | 4.30 | 0.00 | 65.80 | 43.40 | 62.20 | 62.20 | 73.42 |
| Qwen3-Max | 31.37 | 15.44 | 5.26 | 1.23 | 99.17 | 28.90 | 68.90 | 73.23 | **93.70** |
| DeepSeek-V3.2 | 27.50 | 18.64 | **6.50** | 0.00 | **100.00** | 22.29 | 34.40 | 32.30 | 91.54 |
| Qwen3-235B-A22B-Instruct | 17.70 | 9.23 | 4.50 | 0.00 | 97.50 | **21.46** | 50.70 | 58.40 | 79.23 |
| Qwen3-32B | 3.50 | 1.16 | 1.20 | 0.00 | 99.00 | 23.11 | 46.90 | 59.30 | 69.28 |
| Qwen3-14B | 3.30 | 0.87 | 1.39 | 0.00 | 92.50 | 43.99 | 44.35 | 56.60 | 80.50 |
| Qwen3-8B | 2.80 | 0.69 | 1.46 | 0.00 | 80.00 | 40.17 | 51.00 | 53.00 | 80.80 |

credit for covered scoring points, whereas Avg@k requires all answers in a trajectory to be correct. (2) Efficiency. Efficiency is quantified by the total number of tool-invocation steps ($T_{total}$) within a trajectory, reflecting the computational and interaction cost of agent reasoning. (3) Process completion. To measure whether an agent successfully completes the entire reasoning process, we define the Completion Rate (**CR**) as the percentage of tasks in which the agent answers all sub-questions within a predefined limit of 70 tool calls. CR is calculated as $CR = N_{complete}/N_{task}$, where $N_{complete}$ denotes the number of completed tasks, and $N_{task}$ is the total number of tasks.

# 5. Experiments

## 5.1. Experimental Setup

**Baselines.** We conducted a comprehensive evaluation on the TableAgent-Bench, systematically assessing 25 prominent large language models, including the Qwen3 series (Hui et al., 2024a;b; Team, 2025; Yang et al., 2025a;b), Kimi-K2 series (Kimi Team, 2025), DeepSeek-V3.2 (DeepSeek-AI, 2025), Mimo-V2-Flash (Xiaomi, 2026), GPT-5, Claude-4.5-Haiku, Gemini-3-Pro-Preview, MiroThinker-v1.5-30B (Team et al., 2025b), Tongyi-DeepResearch-30B-A3B (Team et al., 2025c), and the GLM series (Team et al., 2025a). Due to their inability to retrieve tables from file systems and insufficient support for long-sequence tool calls and extended context, existing sheet agents were excluded from the baselines.

**Implementation Details.** For fairness, we standardize the system prompt of table-agent baselines(see Appendix D.2). Meanwhile, we implemented a Function Call-based agent architecture (Yao et al., 2023), with all models utilizing their official tool-calling formats. The maximum context length was capped at 32k and the maximum number of tool-call steps was constrained to 70. To reduce results' randomness, we set temperature T=0 and all experiments were repeated 3 times, with metrics averaged to mitigate stochastic effects. More details see Appendix D.4.

## 5.2. Main Results

As shown in Table 4, we conducted performance comparisons across various baseline models. Our findings reveal: (1) **Overall agent performance remains far from reliable.** Across both thinking and non-thinking modes, performance varies substantially across models. Among closed-source models, Gemini-3-Pro-Preview achieves the strongest overall results, attaining the highest Table Relevance Recall (TRR, 85.3%) and Information Coverage (IC, 53.4%). For open-source models, Kimi-K2-Thinking exhibits the best table localization ability (TRR=82.9%), while GLM-4.7 achieves the highest IC (47.4%). Despite these improvements, all models show extremely low end-to-end success rates, with Avg@3 and Avg@3$_{sub}$ capped at 5% and 28.25%. Meanwhile, several leading models achieve near-perfect completion rates, indicating that agents can often finish the dialogue within the tool-call budget but still fail under the strict trajectory-level correctness criterion. This gap sug-

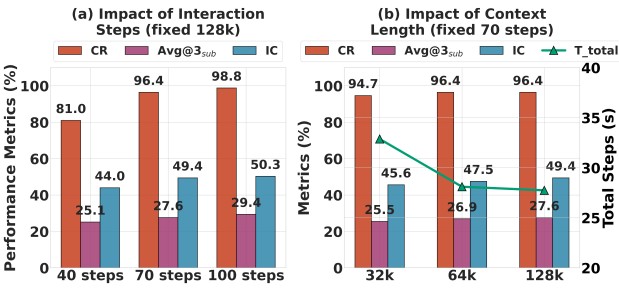

*Figure 5.* Impact of context length and interaction steps on models.

gests that a single retrieval, computation, or propagated reasoning error can invalidate an otherwise completed trajectory, revealing a substantial reliability gap in real-world table-centric analysis. (2) **Reasoning mode and table localization are key performance drivers**. Enabling reasoning mode consistently improves agent performance, particularly in tool utilization and information completeness. For example, Kimi-K2-Thinking improves IC by 12.1% over its non-thinking counterpart. Trajectory-level analysis suggests that reasoning helps agents better interpret tool feedback and preserve global task intent, whereas non-thinking models tend to over-focus on local tool outputs or terminate with incomplete answers. Moreover, table localization accuracy plays a critical role: models with higher TRR and TRP consistently achieve higher IC while requiring fewer tool calls ($T_{total}$), indicating more efficient and directed search strategies. (3) **Model scale and tool call reliability limit multi-table reasoning**. Small models (less than 32B parameters) exhibit limited effectiveness on complex multi-table tasks. Although some achieve moderate table localization accuracy, their IC scores remain below 16%, highlighting substantial room for improvement. In addition, tool call reliability strongly affects trajectory efficiency: models with low tool-call success rates ($R_{success}$) frequently incur extended trajectories due to repeated debugging, which often leads to failure under the 70-step interaction constraint.

### 5.3. Further Analysis

**Context Length Analysis.** To investigate how context length and interaction steps affect tabular agent performance, we conducted systematic experiments. Results in Figure 5 show that increasing steps at fixed 128k context improves performance, as does increasing context length at fixed 70 steps. Notably, CR exhibits a sharp increase initially before plateauing. Considering efficiency-performance tradeoff, we set the maximum interaction step to 70. Meanwhile, since context length has minimal impact on CR, we opt for 32k as agent's context length setting in main experiences.

**ToolSet Ablation Study.** To isolate the impact of different tool categories, we evaluate 4 functional groups: Code Execution (CE), Exact-Matching (EM), Semantic-Retrieval

*Table 5.* Ablation study on tool combinations across DeepSeek-V3.2, Mimo-V2-Flash, and Claude-Haiku-4-5.

| Tool Combination | IC (%)↑ | Avg@3 $_{sub}$ (%)↑ | $T_{total}$ ↓ | TRR (%)↑ |
|---|---|---|---|---|
| **DeepSeek-V3.2** | | | | |
| CE | 43.0 | 25.42 | 40.10 | 65.1 |
| CE+EM | 41.50 | 20.90 | 39.40 | **78.4** |
| CE+EM+SR | 44.90 | 25.42 | 35.33 | 78.2 |
| CE+EM+SR+TP | **46.80** | **27.12** | **35.25** | 74.80 |
| **Mimo-V2-Flash** | | | | |
| CE | 37.50 | 21.33 | 38.31 | 67.30 |
| CE+EM | 43.60 | 24.85 | 32.62 | 79.30 |
| CE+EM+SR | 45.00 | 25.31 | **29.90** | 79.40 |
| CE+EM+SR+TP | **45.60** | 25.48 | 32.87 | **80.10** |
| **Claude-Haiku-4-5** | | | | |
| CE | 46.20 | 25.99 | 43.00 | 68.6 |
| CE+EM | 48.80 | 24.40 | 39.13 | 72.5 |
| CE+EM+SR | **53.40** | **28.25** | **32.80** | **80.6** |
| CE+EM+SR+TP | 51.10 | 24.38 | 36.62 | 76.50 |

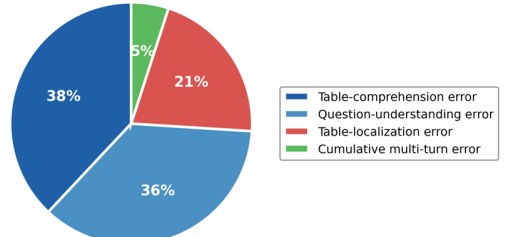

*Figure 6.* Distribution of error cases in TableAgent-Bench.

(SR), and Table Processing (TP). As shown in Table 5, we find: (1) Search-driven tools (EM/SR) are superior to CE. By replacing folder-based browsing with keyword and semantic indexing, EM and SR significantly boost TRR accuracy and interaction efficiency. (2) TP provides general gains but shows diminishing returns on stronger models. While TP generally improves IC and Avg@3$_{sub}$, it brings slight fluctuations on Claude-Haiku-4.5. This may result from functional redundancy and localization noise: strong coding models can often inspect schemas and preprocess tables directly with Python, while extra TP operations may introduce intermediate files that increase candidate-file ambiguity. Thus, TP improves robustness for most models but may slightly hurt models with strong native table-processing ability. More details of tool call analysis see in Appendix B.

**Human Evaluation and Error Analysis.** To validate our metrics, we ask 3 annotators to manually annotate 100 randomly sampled trajectories across multiple dimensions (IC, CQ, and Avg@3). The high agreement with automated scores (Cohen's Kappa = 0.851) confirms the reliability of our evaluation framework. As shown in Figure 6, systematic analysis of failure modes in these trajectories reveals 4 primary error sources: (1) Localization: Ambiguous filenames or deep directory structures hindering table retrieval. (2) Understanding: Incomplete coverage of query perspectives or misinterpretation of domain-specific indicators. (3) Comprehension: Structural complexity of tables leading to repeated code generation failures. (4) Propagation: Cumulative er-

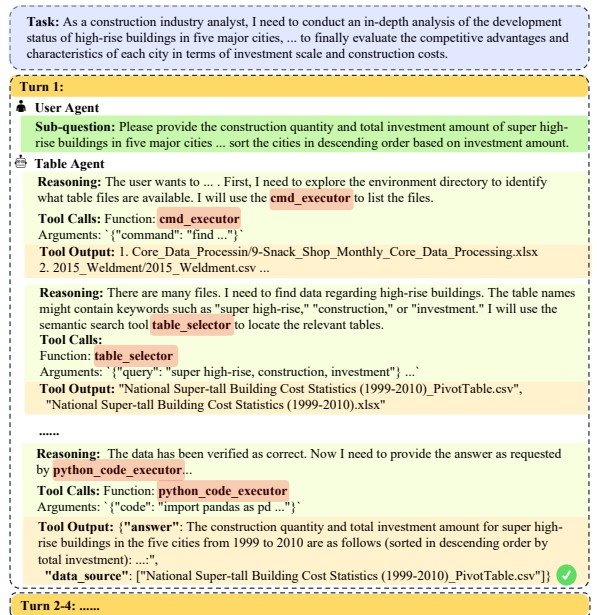

*Figure 7.* Case study of interaction between user/table agents.

rors where early mistakes derail downstream reasoning. Beyond these categories, the failures reveal three underlying bottlenecks: brittle retrieval under vague or cross-lingual indicators, insufficient adaptation to irregular table structures, and myopic debugging that fixes local execution errors without re-checking global task constraints. Detailed examples are shown in Figure 8 and Figure 9.

**Case Study.** The case study result is shown in Figure 7. The Table Agent attempts to address queries from the User Agent through multi-turn tool invocation, first retrieving relevant tables via exact matching or semantic search, then generating and executing Python code for data extraction and computation. While this demonstrates the framework's preliminary capability for handling complex queries involving multi-turn interactions, benchmark evaluations reveal great limitations and persistent errors.

## 6. Conclusion

In this paper, we transition TableQA from passive reading to a proactive interaction paradigm via TableAgent-Bench. Grounded in 2,275 industrial tables, this benchmark utilizes a topology-aware strategy to capture dynamic intent and multi-table dependencies, while the proposed TAEF framework diagnoses intermediate failure modes by evaluating dimensions such as table localization and tool-invocation rationality beyond mere terminal metrics. Our evaluation of 25 state-of-the-art models reveals a substantial capability gap, particularly in maintaining reasoning consistency within long-horizon scenarios. By providing a rigorous testbed and

diagnostic toolset, this work establishes a foundation for advancing autonomous agents capable of robust, real-world table-centric reasoning. While TableAgent-Bench focuses on text-dominant table analysis involving retrieval, structural preprocessing, and programming-based extraction, future extensions can further broaden its scope to multimodal spreadsheets and operation-oriented tasks such as content filling, validation, formatting, and style editing.

## Acknowledgements

This work was supported by the National Natural Science Foundation of China (No. 62176029 and No. 62506050), the China Postdoctoral Science Foundation Funded Project (No. 2024M763867), the Open Competition Program of Chongqing Municipal Commission of Economy and Information Technology (No. YJX-202500100200X), and the Chongqing Science and Technology Innovation and Application Development Projects (No. CSTB 2023TIAD-KPX0064 and No. CSTB 2024TIAD-KPXO154). Moreover, the experimental and computational work in this research partly run on the Huawei Cloud AI Compute Service. We appreciate the stable compute supply from this platform.

## Impact Statement

This work aims to advance the evaluation of LLM-based agents for real-world table analysis. By introducing a benchmark and evaluation framework for multi-turn, multi-table, and tool-based reasoning, it can help researchers and practitioners identify failure modes in table localization, tool use, computation, and long-horizon reasoning. This may support the development of more reliable AI assistants for business analytics, finance, operations, and other data-intensive workflows.

The potential negative impact is that over-reliance on imperfect table agents may lead to incorrect analyses and misleading decisions, especially in high-stakes industrial or financial settings. In addition, tabular data may contain sensitive or proprietary information, requiring careful data governance and privacy protection. Our benchmark is intended for research and evaluation rather than direct deployment validation. Practical use of table agents should include human oversight, secure execution environments, privacy-preserving data handling, and additional checks for numerical and factual correctness.

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

## A. Detailed Error Analysis

To systematically categorize and investigate failure modes, we conduct manual error analysis on the 100 sampled trajectories from different models. As summarized in Table 6, the observed errors can be systematically classified into four categories, with detailed cases shown in Figure 8 and Figure 9. **First**, table localization errors arise chiefly from ambiguous file names, excessively deep directory paths, or semantically vacuous folder names, preventing the agent from retrieving the target table. **Second**, question understanding deviations occur when the agent incompletely covers the multiple perspectives required by the query, or misinterprets domain-specific indicators leading incorrect numerical formulas. **Third**, table comprehension errors stem from structural complexity or ambiguity in the tables themselves, causing repeated failures in code generation and subsequent data extraction. **Finally**, cumulative propagation errors propagate earlier mistaken outputs through the dialogue, precipitating further downstream mistakes.

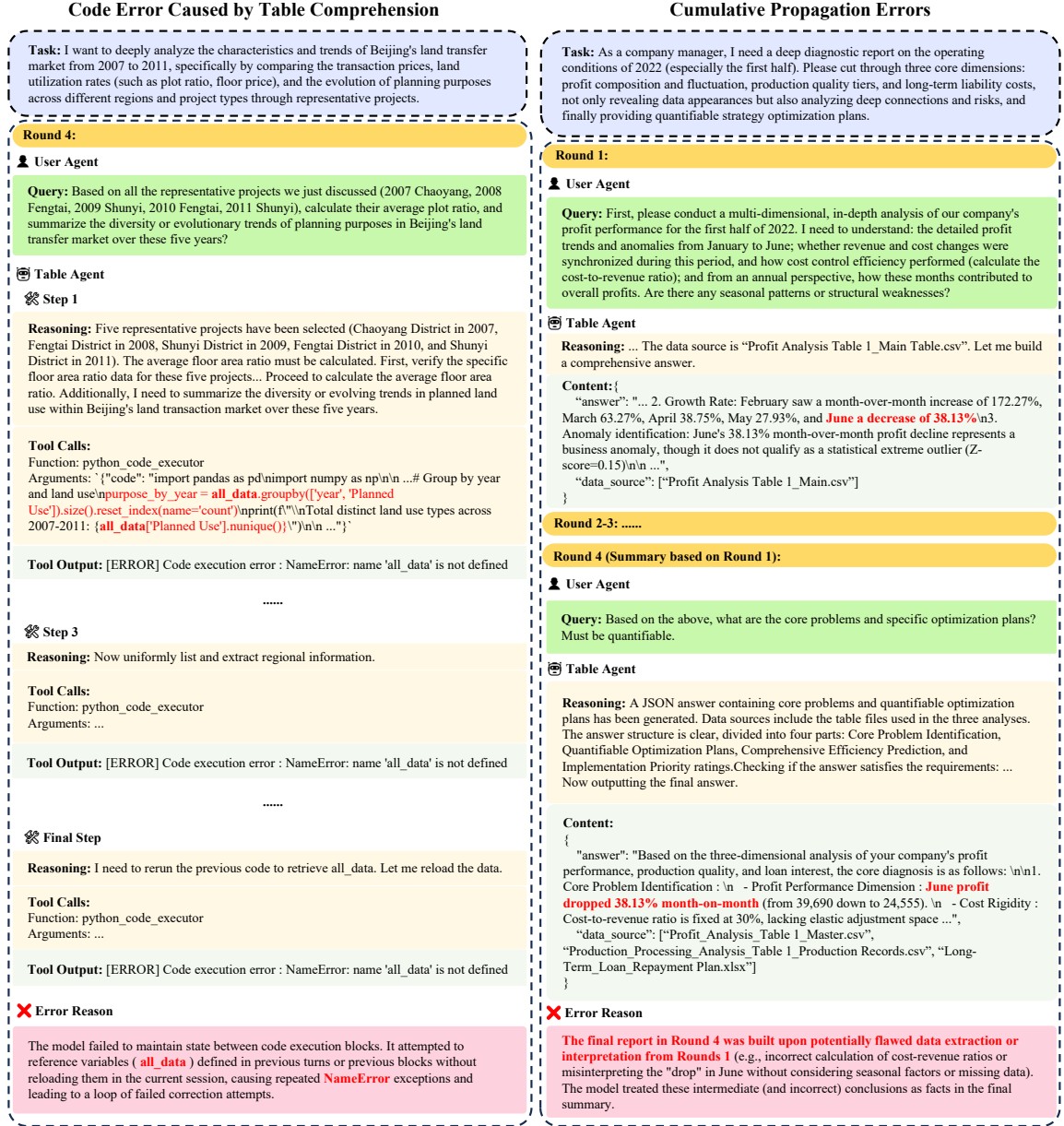

*Figure 8.* Error cases of localization and understanding.

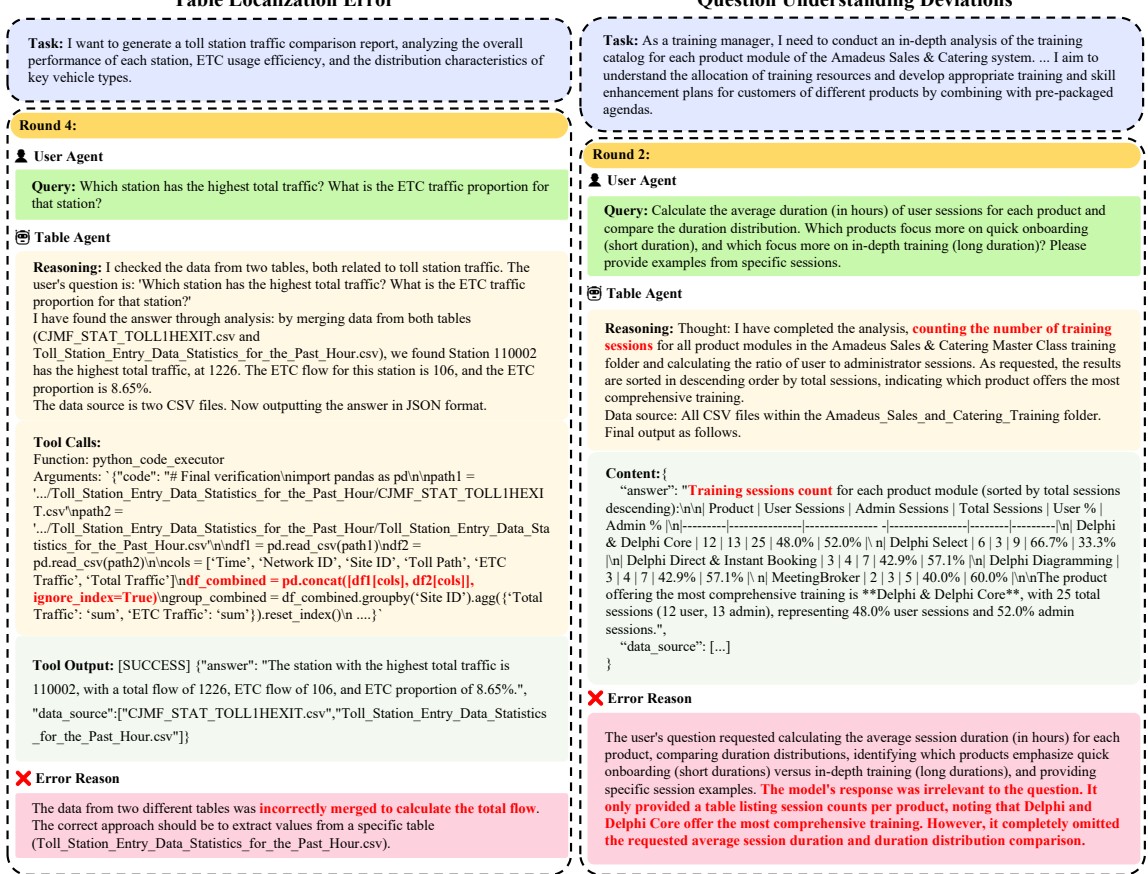

*Figure 9.* Error cases of comprehension and propagation.

## B. More Details for Tool Usage

As shown in Table 6, we summarize the usage statistics of different tools across all experiments. The code execution tool accounts for the highest proportion of tool calls (64.75%), with an average of 20.41 calls per trajectory—significantly exceeding other tool categories. In contrast, table retrieval and table processing tools are used at comparable levels. Further analysis of the interaction trajectories reveals the following insights:

(1) **Limited reuse of retrieval and processing tools due to contextual persistence**. In multi-turn table-based dialogues, individual retrieval and processing tools are typically invoked only once or twice per multi-turn dialogue. This is because sub-questions within a dialogue often depend on shared related tables. Once the relevant table is retrieved and preprocessed in the initial turns, subsequent queries can operate on the same context, eliminating the need for redundant retrieval or reprocessing.

(2) **Iterative debugging enhances table reasoning capability**. The prevalent use of the Python executor stems from table structural complexity and the difficulty of generating correct cross-table analytical code in a single attempt. Models frequently produce erroneous code, necessitating multiple executions and iterative debugging to identify and rectify errors—thereby incurring substantial tool call overhead. Nevertheless, unlike end-to-end one-shot code generation, the Table Agent leverages iterative tool invocations for progressive code refinement, ultimately achieving superior performance in resolving complex table queries.

(3) **Tool call failures stem from code implementation errors**. Despite strong tool invocation capabilities and few parameter inference errors, most tool call failures result from coding mistakes—such as incorrect column indexing and type mismatches—induced by table complexity.

*Table 6.* Tool Call Statistics and Error Analysis

| Tool Name | Tool Call Rate (%) | Avg Tool Calls/Trace | Error Rate (%) |
|---|---|---|---|
| **Code Execution Tools** | **64.57** | **20.41** | **28.66** |
| python_executor | 58.00 | 17.93 | 27.60 |
| cmd_executor | 6.57 | 2.48 | 1.06 |
| **Table Retrieval Tools** | **19.49** | **7.37** | **0.87** |
| table_selector | 6.59 | 2.49 | 0.00 |
| semantic_row_retriever | 5.55 | 2.10 | 0.00 |
| semantic_column_retriever | 2.04 | 0.77 | 0.00 |
| grep_search | 5.31 | 2.01 | 0.87 |
| **Table Processing Tools** | **20.57** | **7.78** | **2.60** |
| table_head_reader | 11.60 | 4.39 | 2.60 |
| xlsx_to_csv_converter | 4.08 | 1.54 | 0.57 |
| header_merger | 2.55 | 0.96 | 0.00 |
| table_decompser | 2.34 | 0.89 | 0.00 |

## C. Implementation Details for Benchmark Construction

### C.1. Data Source of TableAgent-Bench

The tables used for benchmarking are sourced from both publicly available online resources and commercially purchased datasets. The online resources include municipal open data platforms, the official website of the National Bureau of Statistics, industry association portals, and open table datasets, with specific data sources listed in Table 7. Commercially purchased data consists of open-license tables and industry reports obtained from professional data service providers. Additionally, our proprietary collection includes anonymized tables derived from real-world industrial applications.

*Table 7.* The data sources of TableAgent-Bench

| SOURCES | WEBSITES |
|---|---|
| **OPEN-SOURCE DATA PLATFORMS** | |
| WORLD BANK GROUP | HTTPS://DATACATALOG.WORLDBANK.ORG/ |
| NATIONAL BUREAU OF STATISTICS OF CHINA | HTTPS://WWW.STATS.GOV.CN/SJ/ |
| KAGGLE | HTTPS://WWW.KAGGLE.COM/DATASETS |
| CHINA ASSOCIATION OF AUTOMOBILE MANUFACTURERS | HTTP://WWW.CAAM.ORG.CN/ |
| BEIJING PUBLIC DATA OPEN PLATFORM | HTTPS://DATA.BEIJING.GOV.CN/ |
| THE UNITED STATES GOVERNMENT'S OPEN DATA SITE | HTTPS://CATALOG.DATA.GOV/DATASET |
| CHINA SECURITIES REGULATORY COMMISSION DATA PLATFORM | HTTP://WWW.CSRC.GOV.CN/CSRC/TJSJ/INDEX.SHTML |
| SHANGHAI PUBLIC DATA OPEN PLATFORM | HTTPS://DATA.SH.GOV.CN/VIEW/DATA-RESOURCE/INDEX.HTML |
| CELESTRAK | HTTPS://CELESTRAK.ORG/ |
| **TABULAR DATASETS** | |
| MIMOTABLE (LI ET AL., 2024) | HTTPS://GITHUB.COM/JASONNLP/MIMOTABLE |
| WIKITQ (PASUPAT & LIANG, 2015) | HTTPS://GITHUB.COM/PPASUPAT/WIKITABLEQUESTIONS |

### C.2. Domain and Subdomain of TableAgent-Bench

The 6 domains and 28 subdomains in TableAgent-Bench are shown in Table 8.

### C.3. Annotation Team Composition

Our annotation team consists of 25 bilingual annotators, proficient in both Chinese and English, all of whom have passed international language proficiency exams, such as an IELTS score of 6.0 or a CET-6 certification. The majority of team members are native Chinese speakers. Each annotator holds at least a bachelor's degree and has a minimum of one year of experience in data analysis, ensuring high-quality annotation work. The team includes both experienced annotators and

*Table 8.* The 6 domains and 28 subdomains in TableAgent-Bench

| DOMAINS | SUBDOMAINS |
|---|---|
| ENGINEERING SCIENCE | MANUFACTURING AND INDUSTRIAL PRODUCTION; CHEMICAL AND MATERIALS ENGINEERING; ENERGY PRODUCTION AND POWER SYSTEMS; AUTOMOTIVE AND MECHANICAL ENGINEERING; CIVIL ENGINEERING AND CONSTRUCTION; MINING AND RESOURCE EXTRACTION; AGRICULTURAL PRODUCTION AND RURAL ECONOMY |
| SOCIAL POLICY ADMINISTRATION | EDUCATION AND ACADEMIC RESEARCH; GOVERNMENT ADMINISTRATION AND PUBLIC SERVICES; HEALTHCARE SYSTEMS AND PUBLIC HEALTH; SOCIAL SECURITY AND WELFARE PROGRAMS; TRANSPORTATION NETWORKS AND INFRASTRUCTURE; LAND AND WATER RESOURCE MANAGEMENT |
| BUSINESS OPERATIONS | SUPPLY CHAIN AND LOGISTICS MANAGEMENT; RETAIL TRADE AND E-COMMERCE PLATFORMS; MARKETING AND SALES MANAGEMENT; HUMAN RESOURCES AND CORPORATE MANAGEMENT; INTERNATIONAL TRADE AND COMMERCE |
| MACROECONOMICS STATISTICS | ECONOMIC DEVELOPMENT AND FISCAL POLICY; LABOR MARKET AND EMPLOYMENT STATISTICS; TAX POLICY AND REVENUE MANAGEMENT |
| FINANCIAL SERVICES | BANKING AND INSURANCE SERVICES; INVESTMENT AND CAPITAL MARKETS |
| CONSUMER LIFESTYLE | TOURISM AND HOSPITALITY SERVICES; FOOD AND BEVERAGE SERVICES; REAL ESTATE AND HOME IMPROVEMENT; ENTERTAINMENT, SPORTS AND CULTURE; INFORMATION TECHNOLOGY AND CYBERSECURITY |

senior members, the latter possessing strong expertise in table data analysis and annotation. Senior members handle complex tasks efficiently and serve as quality control reviewers, conducting final validations to ensure accuracy and consistency throughout the dataset development process. They also oversee the daily work of annotators, providing technical guidance and addressing any issues that arise during annotation. All annotators work eight hours a day, with an average daily salary of $40. Before beginning work, annotators undergo comprehensive training through online or video sessions, covering annotation guidelines, task requirements, and the academic research applications of the data. This training ensures adherence to standardized annotation procedures. After training, annotators must pass a practical test before commencing official work.

## C.4. Detailed Definitions and Examples of Topology Structures

This paper introduces six topology structures to describe the dependencies and interaction patterns between information points in complex multi-table tasks. Each topology represents a distinct type of reasoning process and task execution path, aiding in the clarification of multi-turn dialogue flow and providing a structured framework for evaluating model reasoning capabilities. The following sections present the definitions, applicable scenarios, and specific examples of these topologies, enabling readers to better understand how to design tasks using these structures and apply them to multi-table reasoning tasks. The topology definitions are as follows:

**1. Chain**
- Definition: Information points strictly depend on each other in a linear sequence, with a progressive relationship, where the next question relies on the context of the previous one.
- Applicable Scenarios: Suitable for tasks that require step-by-step analysis, where each step builds on the result of the previous one.
- Example:
    A = Top five provinces by GDP
    B = Which of these provinces have national-level AI demonstration zones?
    C = Which provincial capitals have the highest number of AI companies?
    D = Which AI company has the highest per capita output?
- Diagram:
    A → B → C → D

**2. Tree**
- Definition: Starting from a root node, the structure branches downward, containing at least two levels; sub-branches can further refine the analysis, but paths within each branch remain independent and chain-like.
- Applicable Scenarios: Suitable for obtaining overall information first, followed by a layered breakdown into specific details, forming a hierarchical expansion.
- Example:

A = Major food crops in China (assumed to be rice and wheat)
B = Major rice-producing provinces (assumed to be Jiangxi and Hunan)
C = Major wheat-producing provinces (assumed to be Henan)
D = Rice yield in Hunan province
E = Rice planting area in Jiangxi province
F = Total wheat production in Henan
- Diagram:
  $A \rightarrow B, A \rightarrow C, B \rightarrow D, B \rightarrow E, C \rightarrow F$

### 3. Fork-Join
- Definition: Multiple information points are independently retrieved in parallel (without dependencies), and later combined into a single comprehensive conclusion
- Applicable Scenarios: Suitable for analyzing the same theme from multiple independent angles, then merging the results into a final conclusion.
- Example:
  A = AI industry scale in Beijing in 2024
  B = AI industry scale in Shanghai in 2024
  C = AI industry scale in Shenzhen in 2024
  D = Combine data from the three cities to recommend the most suitable city for setting up an R&D center
- Diagram:
  $A \rightarrow D, B \rightarrow D, C \rightarrow D$

### 4. Star (One-to-Many)
- Definition: A central information point is queried in parallel across multiple orthogonal dimensions; all sub-questions depend only on the central point, and no further expansion occurs within each sub-path.
- Applicable Scenarios: Suitable for asking multiple questions about the same table or entity from different attribute dimensions, without delving deeper into any one dimension.
- Example:
  A: Query the record with order ID ORD1001
  B: What is the customer's name?
  C: When was the order placed?
  D: What is the total amount of the order?
- Diagram:
  $A \rightarrow B, A \rightarrow C, A \rightarrow D$

### 5. Loop-back
- Definition: During the progression of the main chain, a temporary backtrack is made to an earlier step (usually the starting point) for clarification, correction, or confirmation, after which the analysis continues linearly.
- Applicable Scenarios: Suitable when it is necessary to re-confirm or revise initial assumptions during the analysis.
- Example:
  A = I plan to purchase a car with a loan and need to analyze the impact of different configurations on monthly payments and total interest.
  B = For a car priced at 190,000, with a 30% down payment and a 24-month loan, what are the monthly payment and interest?
  C = If the car price increases to 260,000, with a 35% down payment and an 18-month loan, how will the monthly payment and interest change?
  D = Can you provide an example with no management fee, a higher car price, and a longer loan term?
  E = Based on these cases, how do we calculate the capital cost for each plan, and which one has the highest cost?
- Diagram:
  $A \rightarrow B, B \rightarrow C, C \rightarrow D,$
  $D \dashrightarrow$ Backtrack and revise $A, D \rightarrow E$

### 6. Mesh
- Definition: After multiple branches expand, subsequent queries can freely jump back to historical nodes of different branches, forming cross-path references.
- Applicable Scenarios: Suitable for non-continuous, cross-branch context referencing, simulating the human ability to freely jump between topics.
- Example:
  A: Query total sales data for all categories in Q1 2024
  B: Sales of electronics
  C: Sales of home appliances?

D: Which category of electronics has the highest profit margin?
E: What is the percentage of air conditioners in the home appliance sales mentioned in Q3? (At this point, the Q3 branch has ended, and Q5 actively jumps back to reference Q3)
- Diagram:
   A → B, A → C, B → D, D → E,
   E --→ Cross-branch backtrack $C$

## C.5. Task Templates and Task Design Details

In the design of multi-table reasoning multi-turn dialogue tasks, each task is automatically generated based on the table information from the data folder and employs an appropriate topology to organize the reasoning process. This process includes selecting task templates, randomly choosing topologies, and generating candidate tasks with topological structures. The following is the task generation prompt template:

You are an expert in constructing multi-turn dialogues, specializing in extracting valuable information from complex data to help users design tasks suitable for multi-turn reasoning. When designing tasks with topological structures, you should follow these steps:
## 1. Randomly select 5 task templates:
Randomly choose 5 task templates from the task template list.
## 2. Choose a topology based on the task:
For each task template, randomly select 2 to 4 appropriate topologies and generate tasks suitable for multi-turn reasoning based on the provided table data.
## 3. Output multi-turn reasoning tasks:
Generate multi-turn reasoning tasks based on the task templates and topologies, ensuring each task clearly defines the scenario, purpose, content, and reasoning topology.
Each task should have a clear scenario, purpose, analysis content, and the appropriate topology to ensure a well-defined reasoning path in the multi-turn dialogue.

# Key Concepts
## Task (Core Task, Main Query):
The core goal of the task or the analytical need the user seeks to fulfill, typically involving the task's scenario, analytical objective, and required data.
## Inference Topology:
The data path designed to complete the task, determining the sequence of sub-questions during the analysis process. By selecting the appropriate topology, it helps organize the task's multi-turn reasoning.

# Scenario Design Principles:
## Task Background:
Clarify the background, analytical objective, and scope of the task. Each task description should include background, purpose, and analysis scope.
## Design Thinking:
Follow the topological structure design thinking and define the logical flow of each dialogue round. Every step of the task should clearly demonstrate the data queries and reasoning process.

# Input Data:
## Table Data Summary (Multiple tables are separated by "——")
{table_summaries}

## Task Templates:
Please randomly select 5 task templates from the following list and analyze them with the provided table data:
1.**Comparative Analysis**
Task Description: Compare the differences and similarities across multiple entities, objects, or time periods.
Example Queries:
-Compare the sales growth of different regions between 2020 and 2021, and analyze the discrepancies.
-Compare the market performance of multiple products across different time periods, exploring their similarities and differences.
2.**Trend Analysis**
Task Description: Analyze the trend, growth or decline patterns, and evolution of time-series data.
Example Queries:
-Analyze the annual sales trend of a particular industry over the past five years and forecast the upcoming months.
-Study the yearly temperature trends of a city and uncover its long-term growth or decline patterns.

3.**Distribution Analysis**
Task Description: Analyze the distribution patterns, structural proportions, and ratio relationships of the data.
Example Queries:
-Analyze the proportion of different age groups using a particular service and identify the primary user group.
-Examine the income distribution across regions and assess income inequality.
4.**Causal Analysis**
Task Description: Investigate the causes, influencing factors, drivers, and causal relationships.
Example Queries:
-Analyze the causal relationship between advertising expenditure and sales growth, identifying key influencing factors.
-Examine the relationship between educational investment and student performance, exploring potential drivers.
5.**Evaluation and Diagnosis**
Task Description: Evaluate a situation, diagnose problems, identify risks, and detect anomalies.
Example Queries:
-Assess the financial status of a company and analyze potential financial risks.
-Detect anomalies in product sales data and identify possible causes.
6.**Ranking and Filtering**
Task Description: Rank objects, filter top/bottom values, and identify extremes.
Example Queries:
-Rank products based on sales volume and identify the top 10 highest-selling items from the past year.
-Filter the 5 regions with the lowest sales in 2019 and analyze the underlying causes.
7.**Aggregation and Statistics**
Task Description: Aggregate and compute statistics such as sums, averages, counts, etc.
Example Queries:
-Calculate the average annual income of a city over the past three years and analyze the trends.
-Summarize product sales data to identify the best-selling products and the regions with the highest sales.
8.**Correlation Analysis**
Task Description: Analyze correlations between variables, association rules, and relationships.
Example Queries:
-Analyze the correlation between advertising spending and sales, identifying the optimal balance between investment and returns.
-Investigate the purchasing correlation between different products and explore potential association rules.
9.**Prediction and Inference**
Task Description: Predict future trends, infer outcomes, and make projections based on existing data.
Example Queries:
-Predict the sales volume of a product in the upcoming months, based on historical sales data and market trends.
-Forecast the rainfall patterns for the next quarter based on weather data from previous years.
10.**Comprehensive Reporting**
Task Description: Generate a comprehensive analysis report involving multidimensional analysis.
Example Queries:
-Write a report on the current state and future trends of a particular industry, incorporating market data, policy environment, and competitive landscape.
-Provide a cross-sector report that compares various regions, product categories, and time periods, offering conclusive recommendations.

## Topology Structure Annotation:
Based on the table data and task templates, randomly select 2 to 4 suitable topologies and annotate them for the task:
{topology_definitions}

# Output Format:
Based on the task template, topology structure, and table data, generate tasks suitable for multi-turn reasoning, and annotate the appropriate topological structures.

```
[
   {
      "task": "Analyze sales trends across different regions over the past five years, identify significant growth or decline, and predict market trends for the upcoming months.",
      "design_thinking": "Following the comparative analysis topology, first extract sales data from Table 1, then perform an annual comparison, followed by identifying fluctuation patterns across regions. Finally, combine multidimensional analysis with trend analysis to form sales predictions.",
      "type": "Tree"
   },
   ...
]
```

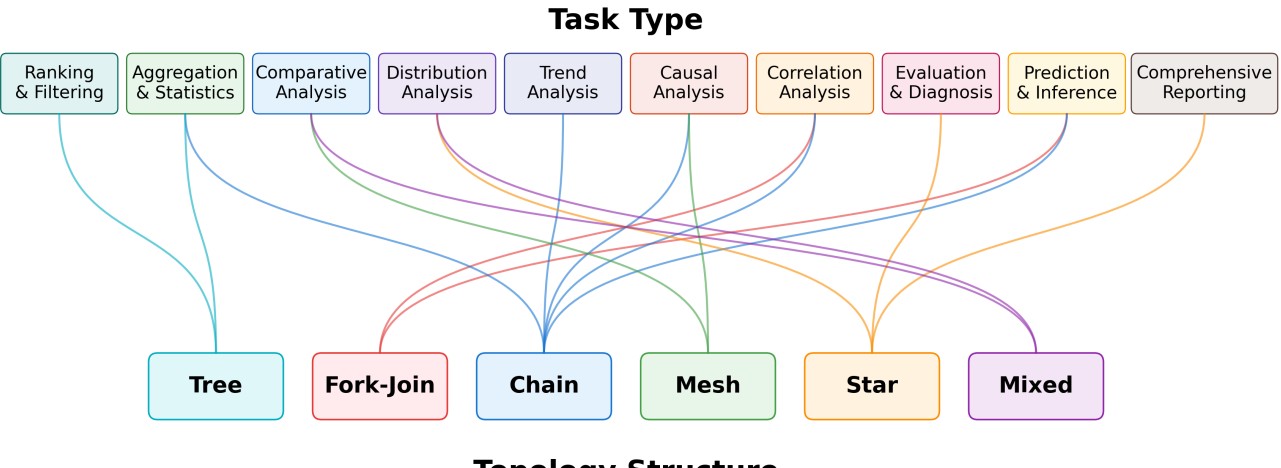

*Figure 10.* The relationship between topology structures and task types.

## C.6. Relationship between Topology Structures and Task Types

Figure 10 illustrates the relationship between topology structures and task types. By analyzing the sample proportions of each topology structure and the distribution of task types, we quantify the association strength between dialogue topologies and task types using the Lift metric. The formula for calculating Lift is as follows:

$$\text{Lift}(T, \tau) = \frac{P(\tau|T)}{P(\tau)} = \frac{\frac{N_{T,\tau}}{N_T}}{\frac{N_\tau}{N}}$$

Where $T$ represents the topology structure type, $\tau$ represents the task type, $N_{T,\tau}$ denotes the number of samples belonging to both topology $T$ and task type $\tau$, $N_T$ is the total number of samples for topology $T$, $N_\tau$ is the total number of samples for task type $\tau$, and $N$ is the total number of samples (1310). We adopt a threshold of 1.2 to filter weak associations and improve interpretability; accordingly, only connections with a Lift value greater than 1.2 are visualized.

The results show that tree and fork-join topologies have the highest sample proportions, at 32.9% and 27.6%, respectively, followed by chain topology at 24.6%. Regarding task types, comparative analysis is the most common, occurring 266 times, followed by comprehensive reporting and trend analysis, with 114 and 83 occurrences, respectively. The strong association analysis reveals that chain topology is most strongly associated with trend analysis, causal analysis, and correlation analysis (Lift > 1.2), while mesh topology shows a significant preference for comparative analysis (Lift = 1.51). Star and mixed topologies demonstrate a marked preference for distribution analysis, with Lift values of 3.47 and 3.58, respectively. Notably, loop-back topology is almost exclusively associated with comparative analysis (Lift = 2.22). Overall, large-sample topologies (e.g., tree and fork-join) exhibit more diverse task type associations, while small-sample topologies show a strong preference for a few specific task types.

## C.7. Dialogue Synthesis

In this study, we propose a topology-aware dialogue synthesis method for generating dialogues in multi-turn reasoning tasks. The process encompasses the steps involved in multi-turn dialogue generation and integrates quality control strategies to ensure that the generated dialogues meet accuracy and naturalness standards.

We adopt a three-stage synthesis method to generate multi-turn dialogues through the following three-phase prompts:

GENERATION_DIALOGUE_PROMPT_STAGE1

# Task Background
The goal is to generate multi-turn dialogues based on Q&A data from a complex task. It is important to note the fundamental difference between Q&A and multi-turn dialogue: while Q&A provides direct answers to specific, clear questions, multi-turn dialogue involves progressively deeper questioning around the user's conversational intent.

In designing multi-turn dialogues, please follow these steps:
1. Define the Dialogue Task (Task): Clarify the overall goal of the user at the start of the dialogue.
2. Design the Information Retrieval Path: Construct a logically structured information retrieval strategy around the topic, specifying the data required at each step and the corresponding answers.
3. Generate the Dialogue Content: Based on the information retrieval path designed above, generate natural multi-turn dialogue.

# Input Data:
## Dialogue Task (Task)
{task}

## Table Summary (Tables are separated by "——")
{table_summary}

# Task Objective
You are required to design a multi-turn dialogue plan based on the provided inputs, referencing the given `tupu_type`. The plan should include three components:

1. Core Task (Task): Refine and rephrase the Dialogue Task (Task) into a natural language description of the user's overall goal at the beginning of the dialogue. The background context and overall purpose should be outlined first, followed by specific details of the information or scope to be queried, while ensuring it aligns with the original task, given topology, and table summary.
2. Information Points List (checkout_list): Decompose the core task into several information retrieval steps, each representing data the user intends to retrieve.
- From the User's Perspective: Describe the information to be obtained from the user's point of view.
- Topological Structure: Organize the information retrieval according to the specified topology, simulating the user's thought process.
- Coverage: Ensure the structure is maintained while covering all the information from the original answer. Information may be omitted to maintain the topological structure.
- Validity: Each "info_item" and "answer" must be derivable from the provided Table Summary.

3. Topology: Define the logical dependencies between the information points.
Ensure the information checkpoints strictly adhere to the chosen topology.

# Topology
{tupu_type}

# Output Format
After reflecting on the above, provide your final answer in JSON format:

```
[
  {
    "task": "Core task description...",
    "design_thinking": "Topology design and construction process...",
    "checkout_list": [
    {
      "idx": 0,
      "info_item": "Description of information point 1...",
      "answer": "Expected answer for this information point..."
    },
    {
      "idx": 1,
      "info_item": "Description of information point 2...",
      "answer": "Expected answer for this information point..."
    },
    ...
```

```
    ],
    "type": "Topology type",
    "graph": "mermaid graph \n ...\n"
  },
  ...
]
```

GENERATION_DIALOGUE_PROMPT_STAGE2

```
# Task Background
The goal is to generate multi-turn dialogues based on Q&A data from a complex task. It is important to distinguish between Q&A
and multi-turn dialogue: while Q&A involves direct responses to individual, clear questions, multi-turn dialogue progressively
explores related questions surrounding the user's conversational intent.

In designing multi-turn dialogues, please follow these steps:
1. Define the Dialogue Task (Task): Clarify the user's overall goal at the beginning of the dialogue.
2. Design the Information Retrieval Path: Construct a logically structured information retrieval plan, specifying the data needed at
each step and the corresponding answers.
3. Generate the Dialogue Content: Generate natural multi-turn dialogue based on the designed information retrieval path.

# Task Requirements
The topology structure path has already been generated. Please proceed with refining the checkpoints while maintaining the
topology. This can involve reordering adjacent checkpoints or incorporating more information from the original answers.
- Compress the checkpoints to 2-6: If checkpoints are too long, keep the topology intact while compressing the checkpoints.
- Diversity of Questions: While restructuring, feel free to rephrase some checkpoints as more varied questions, including filters,
sorting, and aggregation.
- Increased Difficulty: Some checkpoints can be made more complex by introducing composite indicators, numerical reasoning
problems, multi-table reasoning, or comparative analysis.

Finally, optimize the task description, design thinking, and graph information based on the revised checkpoint list.

# Input Data:
## Dialogue Task (Task)
{task}

## Existing Design and Checkpoints
{multi_dialogue_design}

# Output Format
Provide your final answer in JSON format:

[
    {
        "task": "Core task description...",
        "design_thinking": "Topology design and construction process...",
        "checkout_list": [
        {
            "idx": 0,
            "info_item": "Description of information point 1...",
            "answer": "Expected answer for this information point..."
        },
        {
            "idx": 1,
            "info_item": "Description of information point 2...",
            "answer": "Expected answer for this information point..."
        },
        ...
        ],
```

```
        "type": "Topology type",
        "graph": "mermaid graph LR\n ...\n"
    },
    ...
]
```

GENERATION_DIALOGUE_PROMPT_STAGE3

```
# Task Background
The goal is to generate multi-turn dialogues based on Q&A data from a complex task. It is important to note the fundamental
difference between Q&A and multi-turn dialogue: Q&A provides direct answers to specific, well-defined questions, whereas
multi-turn dialogue involves progressively delving into related questions based on the user's conversational intent.

In designing multi-turn dialogues, please follow these steps:
1. Define the Dialogue Task (Task): Clarify the overall goal of the user at the start of the dialogue.
2. Design the Information Retrieval Path: Construct a logically structured approach to information retrieval, specifying the required
data and corresponding answers at each step.
3. Generate the Dialogue Content: Generate multi-turn dialogue based on the designed information retrieval path.

# Task Requirements
The checkpoints are already designed, but I need you to fill in the following fields based on context:
1. Double-check the accuracy of each checkpoint's answer. If it is incorrect, revise it. Ensure that "info_item" and "answer" can be
derived from the provided Table Summary.
2. Based on the table content, assign the associated table(s) to each checkpoint and add a "related_table" field with the table file
name(s).
3. Add a conclusion field: Attempt to answer the user's core task based on all the checkpoints.

# Input Data:
## Dialogue Task (Task)
{task}

## Table Summary (Tables are separated by "——")
{table_summary}

## Existing Design and Checkpoints
{multi_dialogue_design}

# Output Format
Provide your final answer in JSON format:

[
    {
        "task": "Core task description...",
        "design_thinking": "Topology design and construction process...",
        "checkout_list": [
        {
            "idx": 0,
            "info_item": "Description of information point 1...",
            "answer": "Expected answer for this information point...",
            "related_table": ["Table file name 1", ...]
        },
        ...
        ],
        "conclusion": "Attempt to answer task",
    }
]
```

## C.8. Data Augmentation

To enhance the overall fluency and naturalness of the generated multi-turn dialogues, we further improve them through coreference resolution, ellipsis restoration, and paraphrasing. This results in diversity in both structure and content, closely resembling real human user interactions. The prompts for the 3 types of data augmentation methods are as follows:

DATA_AUGMENTATION_PROMPT_V1

You are a professional large model for enhancing multi-turn dialogues. I would like you to help enhance the expression of sub-questions during a multi-turn dialogue process.

# Task Background

Question answering (QA) and multi-turn dialogue have fundamental differences: QA is a direct response to a single, specific question, while multi-turn dialogue involves progressively asking related questions around the user's conversational intent. Therefore, multi-turn dialogue contains the following elements:

1. Defining the conversation topic (Task): The overall intent and objective of the user at the start of the conversation.
2. Designing the information retrieval path: The approach the user constructs around the topic to gather information, including each step's required information and corresponding answers.
3. Enhanced dialogue content: The actual questions and answers posed based on each information step mentioned above.

# Task Requirements

The information retrieval path has already been defined. I would like you to enhance the expression of each information step's question based on this path. In this process, you cannot modify the semantics of each information step; instead, you should generate enhanced expressions that maintain the original meaning.

1. The questions you enhance should be as concise as possible, omitting intermediate indicators that are redundantly included, and only keeping non-redundant query indicators.

* For example: Calculate the production and export volumes of a product, then calculate the export ratio. You only need to ask to calculate the export ratio, as the production and export volumes will naturally be calculated when solving the export ratio problem.
2. For excessive indicators that cannot be combined, you need to think about whether they can be replaced by synthesized indicators that encapsulate the original multiple indicators. The synthesized indicator should semantically include the original indicators and be computable entirely from the original indicators.
3. For trend analysis or similar questions, consider whether they can be replaced with more specific indicators and questions to avoid ambiguity. Based on the enhanced expression above: Please also modify the answers to align with the enhanced question expressions.

Each sub-question's modified expression should be semantically equivalent to the original expression, and the modified answer should be derived entirely from the original answer.

# Input Information

## 1. Multi-turn Dialogue Topic

{task}

## 2. Information Retrieval Approach

{design_{thinking}}

## 3. Original Information Steps

{checkout_list}

# Output Requirements

Please first analyze step by step and finally provide the answer in JSON format:

```
{{
  "checkout_list": [
    {{
      "id": Sub-question Number (idx),
      "info_item": "Enhanced Expression",
      "answer": "Answer to this expression",
    }},
    ...
```

```
    ]
}}
```

## DATA_AUGMENTATION_PROMPT_V2

You are a professional large model for enhancing multi-turn dialogues. I would like you to help enhance the expression of sub-questions during a multi-turn dialogue process.

# Task Background

Question answering (QA) and multi-turn dialogue have fundamental differences: QA is a direct response to a single, specific question, while multi-turn dialogue involves progressively asking related questions around the user's conversational intent. Therefore, multi-turn dialogue contains the following elements:

1. Defining the conversation topic (Task): The overall intent and objective of the user at the start of the conversation.
2. Designing the information retrieval path: The approach the user constructs around the topic to gather information, including each step's required information and corresponding answers.
3. Enhanced dialogue content: The actual questions and answers posed based on each information step mentioned above.

# Task Requirements

The information retrieval path has already been defined. I would like you to enhance the expression of each information step's question based on this path.
1.**Manufacturing condition omission**:Please randomly select 1-2 questions in the multi-turn dialogue and intentionally omit critical limiting conditions (e.g., year, region, specific product model, etc.), making the question expression vague or incomplete.
-Purpose:This vagueness should force the table_agent to be unable to directly answer and instead prompt the user for clarification to confirm the specific conditions.
-Example:The original question is "What is the profit margin for the first quarter of 2023?" which is modified to "What is the profit margin?" (missing the time constraint).
2.**Keep other questions normal**:Except for the 1-2 vague questions selected above, the expressions of the other questions should remain semantically complete, clear, and logically consistent within the conversation.
3.**Natural language style**:Simulate a user who may express themselves imprecisely due to urgency or preset context.

Note: For the modified vague questions, please retain the **correct answer under complete conditions** (i.e., the answer the user originally intended to query with specific conditions).

# Input Information

## 1. Multi-turn Dialogue Topic

{task}

## 2. Information Retrieval Approach

{design$_t hinking$}

## 3. Original Information Steps

{checkout_list}

# Output Requirements

Please first analyze step by step and finally provide the answer in JSON format:

```
{{
  "checkout_list": [
    {{
      "id": Sub-question Number (idx),
      "info_item": "Enhanced Expression",
      "answer": "Answer to this expression",
    }},
    ...
  ]
}}
```

## DATA_AUGMENTATION_PROMPT_V3

You are a professional large model for enhancing multi-turn dialogues. I would like you to help enhance the expression of sub-questions during a multi-turn dialogue process.

# Task Background

Question answering (QA) and multi-turn dialogue have fundamental differences: QA is a direct response to a single, specific question, while multi-turn dialogue involves progressively asking related questions around the user's conversational intent. Therefore, multi-turn dialogue contains the following elements:

1. Defining the conversation topic (Task): The overall intent and objective of the user at the start of the conversation.
2. Designing the information retrieval path: The approach the user constructs around the topic to gather information, including each step's required information and corresponding answers.
3. Enhanced dialogue content: The actual questions and answers posed based on each information step mentioned above.

# Task Requirements

The information retrieval path has already been defined. I would like you to enhance the expression of each information step's question based on this path. In this process, you cannot modify the semantics of each information step; instead, you should generate enhanced expressions that maintain the original meaning. 1. **Persona creation**: Please assume the persona of someone who speaks very concisely, with a direct and efficient questioning style, avoiding redundant embellishments.
2. **Introducing references and omissions**: In the sub-question expressions of the multi-turn dialogue, introduce references (e.g., "it," "that," "the former") and omissions (e.g., omitting subjects or known conditions) as much as possible to make the dialogue more aligned with natural human conversational habits.
3. **Increasing reasoning difficulty**: Through the use of references and omissions, increase the contextual dependence and reasoning complexity for the table_agent, forcing it to rely on previous dialogue information to accurately understand the intent of the current question.

Each sub-question's modified expression should be semantically equivalent to the original expression, and the modified answer should be derived entirely from the original answer.

# Input Information

## 1. Multi-turn Dialogue Topic

{task}

## 2. Information Retrieval Approach

{design_thinking}

## 3. Original Information Steps

{checkout_list}

# Output Requirements

Please first analyze step by step and finally provide the answer in JSON format:

```
{{
  "checkout_list": [
    {{
      "id": Sub-question Number (idx),
      "info_item": "Enhanced Expression",
      "answer": "Answer to this expression",
    }},
    ...
  ]
}}
```

## C.9. Automated Evaluation

The automated evaluation tool performs a step-by-step assessment of each dialogue sample, primarily focusing on the correctness of sub-questions, table dependencies, and difficulty level. The evaluation process includes:

- Sub-question Level Evaluation: Each sub-question is evaluated for correctness, table dependency, and difficulty, ensuring that the question is executable and not overly simplistic or unsolvable.

- Global Task Level Evaluation: It ensures that the sub-questions in the dialogue cover the expected cross-table retrieval, alignment, and summarization paths, and that each answer aligns with the global task, maintaining logical consistency.

The prompts for sub-question and global task evaluations are as follows:

SUB_QUESTION_EVAL_PROMPT

You are a multi-turn dialogue evaluation expert. Your task is to assess the **correctness of the answers** and **the reasonableness of table dependencies** for each sub-question in the multi-turn dialogue plan.

# Input Data Description
1.**Table Overview and Content**: Full content of the tables involved in the questions.
2.**Multi-turn Dialogue Plan**:checkout_list: A detailed list of sub-questions, including the "answer" (expected answer) and "related_tables" (tables involved).

# Task Description
Please analyze each sub-question in the checkout_list as follows:
1.**Table Dependency Reasonableness (table_dependency)**: Verify whether the listed related_tables are necessary to answer the sub-question and whether all tables mentioned are included in related_tables.
2.**Sub-question Difficulty Rating (type_difficulty)**:
-0: Simple query.
-1: Sub-question includes conditions such as filtering, sorting, or aggregating, or it is a standard multi-table query.
- 2: Involves indicator reasoning or computational inference. Indicator reasoning refers to queries where the required metric does not exist directly in the table and must be derived by combining multiple indicators; computational inference involves deducing conclusions by performing calculations on table data.
- 3: Complex multi-table reasoning. Involves reasoning across multiple tables, based on the indicators and calculations described above.
3. **Answer Correctness (correctness)**: Compare the sub-question's answer with the Table Content Overview and assess its accuracy.

# Input Data
## 1. Table Overview and Content (tables separated by "——")
{table_summary}

## 2. Multi-turn Dialogue Plan (Checkout List)
{multi_dialogue_design}

# Output Format
Please output in the following JSON format:
{
   "checkout_list": [
   {
     "id": Sub-question ID (idx),
     "table_dependency": { "reason": "Brief explanation", "value": "True/False/Unknown" },
     "type_difficulty": { "reason": "Brief explanation", "value": 0/1/2/3 },
     "correctness": { "reason": "Explanation of reasoning, specifying which key point or table the original answer is based on and how the inference is made.", "value": "True/False/Unknown" }
   },
   ...
    ]
}

TASK_EVAL_PROMPT

You are a multi-turn dialogue evaluation expert. Your task is to assess the **overall quality** and **coverage** of the multi-turn dialogue plan. Assume that the answers to sub-questions and table references are correct, and focus on evaluating the following dimensions.

# Input Data Description
1. **Table Overview and Content**: Full content of the tables involved.
2. **Multi-turn Dialogue Plan**: Includes task description (task), design thinking (design_thinking), query path (query_path), and

sub-question list (checkout_list).

# Evaluation Dimensions
1. **Multi-table Reasoning Difficulty Rating (total_difficulty)**: Calculate the ratio of multi-table reasoning questions to the total number of questions in the sub-question list.
2. **Task Coverage in Multi-turn Dialogue (task_coverage)**:True/False: Whether the information retrieved by the sub-questions fully covers the goal defined in the task.

# Input Data

## 1. Table Overview and Content (tables separated by "——")
{table_summary}

## 2. Multi-turn Dialogue Plan
{multi_dialogue_design}

# Output Format
Please output in the following JSON format:

{
    "total_difficulty": { "reason": "Brief explanation", "value": "Number of multi-table questions/Total number of questions" },
    "task_coverage": { "reason": "Brief explanation", "value": "True/False" }
}

## C.10. Manual Review

In the human review stage of the quality control process, experts manually audit the filtered dialogues to correct any remaining errors. This stage is critical for ensuring the accuracy, reliability, and overall quality of the final dataset. The annotation process is structured around several key dimensions, each essential for evaluating dialogue quality:

1. **Accuracy of Relevant Table Paths and Dependencies Between Questions and Tables:** Experts verify that the correct tables are used throughout the dialogue and that the dependencies between questions and tables are properly maintained. This ensures that each question is supported by the relevant table(s) and that the referenced data sources are both appropriate and accurate.

2. **Consistency Between Tasks and Multi-Turn Dialogues:** This dimension assesses the alignment between the specified task and the structure of the multi-turn dialogue. Experts ensure that the dialogue progresses logically, with each question and answer contributing to the completion of the overall task. If the dialogue diverges from the intended task or fails to address the task effectively, corrections are made.

3. **Clarity of Sub-questions and Adherence to the Specified Topology:** Sub-questions are evaluated for clarity, ensuring that each is concise, clear, and unambiguous. Experts also ensure that the sub-questions follow the defined reasoning topology, which dictates the logical flow and dependencies. If a sub-question is unclear or deviates from the expected structure, it is revised to improve clarity and alignment with the topology.

4. **Factual Correctness of Answers:** The factual accuracy of answers is rigorously examined to ensure that each response is supported by the relevant data and is free from errors. Experts confirm that the information in the answers aligns with the provided tables and data, checking for any inaccuracies or unsupported conclusions.

Given the open-ended nature of multi-turn dialogue responses, a rubric-based evaluation protocol is implemented to ensure a structured and consistent approach to manual annotation. Experts are provided with reference answers for key data points, which serve as benchmarks for evaluating the accuracy and completeness of the model-generated responses. The rubric includes specific scoring criteria for critical data and logic, ensuring consistent and transparent dialogue quality assessment.

Scoring point annotation is a critical component of the manual review process. It involves marking the key data and core logic used in the answers to assess the overall quality of the responses. Scoring points focus on the accuracy and relevance of the information provided, as well as the clarity and correctness of the reasoning. For instance, scoring points may include:

- Correctness of comparisons (e.g., comparing traffic volume across regions),

- Accurate use of data points (e.g., ensuring correct numbers from source tables),

- Valid analysis of trends or correlations (e.g., ensuring trends are accurately described and supported by data),

- Appropriateness of derived conclusions (e.g., ensuring conclusions logically follow from the data).

These annotations are vital for evaluating whether model-generated responses meet the required standards and align with the intended task objectives. By carefully reviewing and annotating these key points, experts ensure that the final dataset reflects high-quality dialogue content that accurately captures the reasoning and analysis required for multi-turn tasks.

After manual annotation, the dialogues undergo a final validation process. This ensures that all issues identified during the review are addressed and that the dialogues meet the required quality standards. The revised dialogues are validated for accuracy and consistency, ensuring the dataset is ready for use in subsequent stages of model training, evaluation, and research.

In summary, scoring point annotation is an essential part of the quality control process. It ensures that multi-turn dialogues are accurate, logically consistent, and aligned with task objectives. By combining automated evaluation with expert manual review, we create a robust and reliable dataset suitable for evaluating model performance in complex multi-table reasoning tasks.

## D. Evaluation Details for Table Agent Framework

### D.1. The Defination of ToolSet

The Table Agent in this study integrates a suite of tools to facilitate the retrieval, processing, and analysis of tabular data. These tools are categorized into the following three classes based on their core functionalities:

#### D.1.1. TABLE RETRIEVAL TOOLS

These tools are designed to locate target tables from the file system, supporting both keyword-based exact matching and semantic-based fuzzy retrieval.

- `grep_search`: A text search tool based on regular expressions. It rapidly locates relevant table files by searching for keywords or metrics related to the user's query within specified files or directories.

- `table_selector`: A semantic vector-based tool for table file retrieval. It utilizes the Qwen3-Embedding-0.6B model to encode the name, header (first 10 rows), and footer (last 10 rows) of each table into high-dimensional vectors. The agent generates a search query based on the user's intent to identify the most semantically relevant tables in the vector space. This tool is particularly effective in scenarios involving poorly named files or disorganized directory structures.

- `semantic_row_retriever`: A row-level semantic content retrieval tool. It supports content-based fuzzy queries by encoding each data row along with its corresponding column name into a vector. This tool can accurately locate relevant table rows when user queries involve derived metrics or abstract concepts not explicitly present in the data.

- `semantic_column_retriever`: A column-level semantic retrieval tool. It encodes the column names and the table name of each table into vectors. Its application is similar to that of `semantic_row_retriever`, focusing on responding to fuzzy queries by leveraging the semantics of column definitions.

#### D.1.2. TABLE PROCESSING TOOLS

These tools are responsible for preprocessing and parsing structurally diverse table files, converting them into a standardized format amenable to analysis.

- `xlsx_to_csv_converter`: A table format conversion tool that converts Excel (`.xlsx`) files into the CSV format. During the conversion, it automatically handles files with multiple worksheets and unmerges merged cells to ensure a standardized output.

- `table_reader`: A table content preview tool. It reads and displays a specified number of rows from a table file, allowing for a quick inspection of its content.

- `header_merger`: A multi-level header merging tool. It is designed to process CSV files with complex hierarchical headers (e.g., horizontally or vertically merged cells). It flattens the multi-level header into a single row using a designated separator while preserving the original hierarchical information.

- `table_decomposer`: A complex table parsing tool. It leverages the understanding capabilities of a Large Language Model (LLM) to intelligently parse non-standard or structurally complex tables (e.g., those with multiple headers or interleaved data blocks). It redefines the table's header and data structure, transforming it into a standard two-dimensional format.

### D.1.3. CODE EXECUTION TOOLS

These tools provide fundamental file system operations and code execution capabilities, forming the foundation for the agent to perform complex tasks.

- `cmd_executor`: A command-line execution tool. It allows the agent to execute standard command-line instructions within the operating environment for basic operations such as file management.

- `python_executor`: A code interpreter for executing Python code in an isolated environment. The Table Agent primarily utilizes this tool for in-depth data analysis, such as performing complex data querying, cleaning, transformation, and computation tasks using libraries like `pandas` to derive the final analysis and response.

### D.2. System Prompt of Table Agent

We present a structured system prompt architecture specifically designed for tabular data analysis agents. This prompt framework explicitly integrates five essential components: role definition and task objectives, execution environment constraints, output specifications, current query context, and dialogue history. Significantly, our approach deliberately eschews domain-specific rule-based guidance traditionally employed in table analysis systems. This design choice stems from the demonstrated capability of contemporary large language models to autonomously execute the complete analytical workflow—automatically identifying relevant tables, processing and semantically interpreting tabular content, and generating accurate final responses. Empirical validation reveals that this rule-free paradigm achieves performance comparable to sophisticated domain-rule-enhanced methodologies, while simultaneously offering superior system simplicity and adaptability across diverse analytical scenarios. This finding suggests that explicit domain policies may be unnecessary for high-performing table analysis agents when equipped with appropriate structural prompting. The explicit integration of the current query and dialogue history into the system prompt mitigates user's query truncation risks caused by context window length constraints in large language models.

---

You are a professional tabular data analysis expert. Please strictly follow the procedures and rules below to accurately respond to user queries regarding tabular data.

# 1. Role and Core Objective
- **Role**: A professional tabular data analysis agent, proficient in table preprocessing, tool orchestration, and pandas programming.
- **Objective**: Extract accurate information from tables and answer user questions through rigorous reasoning and correct tool call.
- **Stateless Tools**: The code executor is stateless—each execution is independent and does not retain results from previous runs!
- **Reasoning + Parallel Tool Use**: You must thoroughly reason before invoking any tool, and maximize parallelism in tool calls to accelerate problem solving.

# 2. Task Environment
- **Note: All your operations must be performed within the environment directory using absolute paths.**
- **Current working environment path**: {enviroment}.
Always use full absolute paths for all operations to avoid path errors. File read/write or other operations outside the /tmp directory are strictly prohibited.

# 3. Output Requirements
- **Output Format**: Every response must follow the "Thorough Reasoning + Action" structure. The action can either be a direct

---

answer or a tool call. Answers should be concise and directly present key data and conclusions.
The final answer must be output in a fixed JSON format with two fields: answer provides the response, and data_source lists the source table(s):
```
{
    "answer": "A textual answer addressing the user's question.",
    "data_source": ["table_name_1", "table_name_2", ...]
}
```

# 4. Current Question
{query}

# 5. Conversation History
{conversation_history}

## D.3. The Details of Content Quality Metrics

Traditional evaluation frameworks for open-domain question answering (QA) typically center on dimensions such as accuracy, completeness, and redundancy (). The task of Table Question Answering (Table-QA), however, presents a unique set of challenges. In this context, users expect not only a precise final answer but also a detailed reasoning process to validate its correctness. Consequently, a high-quality response must often integrate complex data analysis, derivation of metrics, and conclusive summarization.

We argue that solely assessing Information Coverage (IC)—that is, the extent to which key information is recalled—is insufficient for this task. We therefore propose two additional dimensions to evaluate the Content Quality(CQ) of a response: **Answer-Internal Consistency** and **Task-Irrelevant Redundancy**.

1. **Answer-Internal Consistency**: This dimension assesses the logical coherence of the generated text itself, examining whether it contains self-contradictory statements or conclusions. The introduction of this metric is crucial to penalize models that attempt to speculatively inflate their IC scores through "enumerative responses" (e.g., "The answer could be A, but it might also be B"), or those that generate conflicting conclusions, thereby confusing the user.

2. **Task-Irrelevant Redundancy**: This dimension measures the alignment of the response content with the user's core intent. By evaluating the presence of data or analysis extraneous to the task, we can indirectly gauge the model's depth of intent comprehension and the precision of its output.

The detailed prompt is as follow:

# Role
You are a meticulous AI evaluation expert responsible for scoring the final response from a Table Agent on a 5-point scale across two dimensions.

# Core Principles
- This evaluation does **not** compare against a "ground truth" answer. It focuses solely on the **internal quality** of the response itself—its logical integrity and informational efficiency.

# Input Data
- **User Query**: The original user query.
- **Model Answer**: The final response text generated by the Table Agent.

# Scoring Dimensions

1. Answer-Internal Consistency
*Definition: Assesses the logical coherence and unity of the response text. This dimension ignores factual correctness and focuses exclusively on whether the answer is self-consistent.*
* **1/5: Severely Contradictory** - The response contains direct and central self-contradictions (e.g., stating "sales increased" in the analysis but "sales decreased" in the conclusion).

* **2/5: Logically Disjointed** - The reasoning provided does not sufficiently support the final conclusion, indicating a significant logical gap.
* **3/5: Ambiguous or Evasive** - The response avoids a definitive conclusion by enumerating multiple possibilities, leading to an uncertain outcome, or there is a slight logical leap between the reasoning and the conclusion.
* **4/5: Largely Coherent** - The main logic is sound, but there may be a minor, non-critical logical flaw or ambiguous phrasing that does not impact the overall understanding of the core conclusion.
* **5/5: Perfectly Coherent** - The logical chain is complete and rigorous. All statements, reasoning, and conclusions are fully aligned, with no internal contradictions or ambiguities.

**2. Task-Irrelevant Redundancy**
*Definition: Assesses whether the response is concise and focused, and whether it contains extraneous information beyond the scope of the user's core question. Redundancy is categorized as either Data Redundancy or Intent Redundancy.*
* **1/5: Severely Redundant** - **Data Redundancy**: Outputs the entire or a majority of the source table (i.e., a "table dump"). **Intent Redundancy**: Completely misinterprets user intent, providing a lengthy, off-topic analysis.
* **2/5: Notably Redundant** - **Data Redundancy**: Outputs a large, unfiltered subset of data, much of which is irrelevant. **Intent Redundancy**: Provides a tangential analysis that is lengthy and distracts from the core answer.
* **3/5: Slightly Redundant** - **Data Redundancy**: Includes minor, non-essential data (e.g., extra columns). **Intent Redundancy**: Makes a slight but unnecessary extension of the user's intent.
* **4/5: Minimally Redundant** - The response is focused but includes negligible extraneous information, such as an unnecessary index column or a trivial formatting artifact.
* **5/5: Perfectly Focused** - The response is exceptionally concise. All information directly serves to answer the user's core question, with no superfluous data, analysis, or formatting.

# Output Requirements (Strict JSON Format)
For each dimension, first provide a detailed reason, then provide the integer score from 1-5.

```
{
  "internal_consistency": {
    "reason": "Explain any logical contradictions, ambiguities, or evasiveness,
               citing specific text from the answer.",
    "score": "Integer 1-5"
  },
  "task_irrelevant_redundancy": {
    "reason": "Specify the type of redundancy (Data/Intent), explain why it is
               task-irrelevant, and locate it within the response.",
    "score": "Integer 1-5"
  },
  "overall_comment": "Summarize the core quality issue in one sentence, e.g.,
                     'The response is logically sound but highly redundant...'"
}
```

# Inputs
1. **User Query**
{query}

2. **Model Answer**
{model_answer}

## D.4. Implementation Details.

We ensured evaluation fairness by standardizing the system prompt for all table-agent baselines (see Appendix D.2). We implemented a Function Call-based agent architecture(Yao et al., 2023), with all models utilizing their official tool-calling formats. Moreover, the maximum context length was capped at 32k tokens with a sliding window mechanism that discards oldest information when content exceeds the limit. Closed-source models were evaluated via official APIs while open-source models were deployed on 8 × A800 GPUs. To guarantee result determinism and reproducibility, we fixed the random seed at 42, set temperature T=0 and top-p=0.95 . The agent-environment interaction was constrained to a maximum of $S_{\max} = 70$ tool-call steps to avoid infinite agent execution. Upon reaching this threshold, agent terminated task execution and disregarded all subsequent queries. All experiments were repeated 3 times, and the metrics were averaged to mitigate stochastic effects.

**Manual Review** The automated evaluation tool performs a step-by-step assessment of each dialogue sample, primarily

focusing on the correctness of sub-questions, table dependencies, and difficulty level. The evaluation process includes:

- Naturalness: Evaluating whether the language used in the dialogue maintains the fluidity typical of natural conversation, avoiding awkward phrasing and unnecessary repetition.

- Coherence: Ensuring that each dialogue turn logically connects with the previous one, with responses supporting each other and preventing disjointedness or information gaps.

- Task Consistency: Verifying that all dialogue turns align with the overall task objective and that the conclusions drawn are accurate.

This structured approach ensures the effective generation of multi-turn dialogues while maintaining high quality and consistency with the task objectives. It also provides essential support for evaluating the model's multi-turn reasoning capabilities and tool-use strategies.

