# OpenReview forum: "How Far Can LLM Agents Reason with Tables? Benchmarking Multi-Turn Agentic Table Question Answering in the Wild"
_ICML.cc/2026/Conference — ICML 2026 regular_

### Official Review · Reviewer_zfRw · 2026-02-27

**Soundness:** 3
**Presentation:** 3
**Significance:** 2
**Originality:** 2
**Overall Recommendation:** 3
**Confidence:** 4

**Summary:**

The paper introduces TableAgent-Bench, a large-scale bilingual benchmark designed to evaluate LLM agents on their ability to perform proactive, multi-turn reasoning within complex, multi-table industrial environments. Unlike previous benchmarks that treat table-based question answering as a passive, single-turn task, this work utilizes a topology-aware construction strategy to simulate realistic human-like workflows across 1,310 dialogues grounded in 2,275 complex tables. To rigorously assess these agents, the authors also propose the TAEF, which includes a specialized toolset and four diagnostic metric categories to identify specific failure modes in the reasoning process

**Compliance With Llm Reviewing Policy:**

Affirmed.

**Final Justification:**

The rebuttal was reasonable; however, I believe that these points are not sufficiently reflected in the current version of the paper. It would be beneficial to address these issues and submit the revised work to the next conference.

**Key Questions For Authors:**

See Weaknesses.

**Limitations:**

The paper does not explicitly state the limitation. Please provide the limitations of the work and the future plan for addressing the limitations.

**Strengths And Weaknesses:**

Strengths

A primary strength of this research is its focus on structural and conversational complexity, moving beyond single tables to include nested hierarchies and interdependent sub-questions that reflect real-world business intelligence scenarios. The implementation of six distinct reasoning topologies (e.g., Chain, Tree, Fork-Join) provides a structured and diverse way to evaluate how information flows across multiple turns, offering a more nuanced assessment than simple terminal accuracy. Furthermore, the paper provides a comprehensive evaluation of 25 SOTA models, revealing a significant capability gap even in leading models like Gemini-3-Pro-Preview, which highlights the benchmark's effectiveness as a challenging and necessary testbed for the next generation of autonomous table agents.

Weaknesses

1. Lack of Detailed Explanation Regarding the Data Construction Process: The paper lacks a comprehensive explanation of the data construction process. While Sections 3.1 and 3.2 mention that tables were collected from public and commercial sources, there is no specific detail regarding which websites were used for extraction or the precise step-by-step procedures for data validation. Although the authors state the dialogue structures were "inspired by human cognitive patterns in multi-table analysis", the paper fails to specify exactly which patterns were observed or how they were translated into the benchmark. Furthermore, regarding Quality Control, the paper mentions that "experts" manually audited the dialogues, yet it provides no definition or background on who these people are or what qualifies them as experts for this specific task.

2. Limitations of the Multi-Turn Setting and Differentiation from Sequential Single-Turn QA: In a realistic multi-turn setting, the agent's previous responses should actively influence the user's subsequent questions. However, the current setting uses pre-constructed dialogue skeletons, which raises questions about whether this truly simulates a real-world conversational flow. If the dialogue is predetermined, how does this differ from simply concatenating several single-turn QA pairs? Please clearly define the distinction between this benchmark's "topology-aware" multi-turn setting and a standard sequence of independent single-turn queries, especially given that real-world intent should evolve dynamically based on agent feedback.

---

> ### Author Rebuttal · Authors · 2026-03-31
>
> Dear Reviewer zfRw,
>
> Thanks for taking the time to review this paper and we appreciate the positive feedback and your detailed questions regarding data construction. We would like to address your concerns as follows:
>
> **Q1: More comprehensive details are required during data construction, including data sources and validation steps, the specific human cognitive patterns used for dialogue design, and qualifications of annotators.**
>
> **A1：** Thanks for this helpful comment. While Appendix C provides construction details, we agree these should be more prominent in the main text. To address your questions: (1) public data sources are listed in Appendix C.1 (Table 7); (2) expert annotators' qualifications are detailed in Appendix C.3 (Line 818); (3) the six topology structures are abstracted from real-world multi-turn tasks (Section 3.2), with task-topology in Appendix C.6 (Line 1067) and dialogue synthesis procedure in Appendix C.7 (Line 1092). We will incorporate a concise summary into the revised main text.
>
> **Q2：The authenticity of the pre-constructed dialogue skeletons are questions and request a clear differentiation between the benchmark's "topology-aware" multi-turn setting and a standard sequence of independent single-turn queries.**
>
> **A2:** Thanks for raising this critical question. The fundamental distinction between our "topology-aware" multi-turn setting and simply concatenating single-turn QA lies in: the dialogue skeletons encode explicit logical dependencies, rather than sequential arrangement of independent queries. This is reflected in:
>
> 1. Topological Dependencies vs. Independent Queries: In standard single-turn QA, $Q_2$ can be answered independently without information from $Q_1$. In contrast, our benchmark encodes explicit topological dependencies, such as chain dependencies (later questions depend on earlier answers), tree dependencies (multiple subsequent questions depend on prior responses), and so on.
>
> 2. The Trade-off between Deterministic Evaluation and Unbounded Dynamic Exploration: While we thoroughly considered adopting fully dynamic agent-user interactions and faced a fundamental design trade-off. Unlike standard task-oriented dialogues (e.g., flight booking) that converge to a verifiable goal, comprehensive data analysis is inherently open-ended. An abstract task like "Analyze German exported cars" holds unbounded analytical freedom (e.g., exploring price trends, geographic distribution, or sales volume). If we allow fully dynamic interactions, the conversational flow would branch into infinite plausible trajectories, rendering objective, automated evaluation virtually intractable. To guarantee a precise assessment of the agent's core tool-use and table-processing capabilities, we made a deliberate design trade-off by utilizing topology-aware static skeletons. Importantly, by mapping various analytical perspectives into diverse topological structures, we still successfully simulate the dynamic branching of human cognition and preserve multi-turn diversity, all while ensuring execution ground truths remain strictly well-defined.
>
> 3. Intra-Turn Complexity and Agentic Reasoning: Beyond inter-turn topological dependencies, each sub-question itself possesses sufficient analytical depth—encompassing cross-table queries, multi-intent combinations, and comprehensive comparative analyses, rather than simplistic lookups. Therefore, even within a single turn, agents must autonomously perform task planning and decomposition, demonstrating that predefined skeletons do not diminish the demand for advanced reasoning capabilities.
>
> **Q3：Lack limitation description.**
>
> **A3:** Considering the current dataset modality scope, we plan to further expand to multimodal settings (e.g., table–image scenarios) in future work. Moreover, specialized tools and processing mechanisms for multimodal table understanding remain underexplored.
>
> **A Broader Perspective on Our Motivation and Insights**：
>
> While addressing reviewers' questions, we emphasize that TableAgent-Bench is more than a dataset; it is a complete evaluation infrastructure for industrial-scale table agents. Our contribution include (1) an industrial-grade toolset for messy real-world tables, (2) end-to-end scenarios requiring autonomous search, understanding, and coding over large directories, and (3) actionable insights from fine-grained trajectory analysis.
>
> Crucially, our benchmark exposes two fundamental barriers to real-world agent deployment.First, unlike traditional TableQA with pre-retrieved tables, the main industrial bottleneck lies in autonomous directory navigation. Second, although models achieve moderate partial success (IC), their low complete-trajectory performance (Avg@3) reveals limited long-horizon reasoning and error recovery, making human auditing costs undermine automation gains. These findings provide a practical roadmap for future agent training, such as domain-specific RL.  More details show in experience section.

---

> > ### Author Rebuttal · Reviewer_zfRw · 2026-04-03
> >
> > Thank you for the detailed response. I would like to see a more thorough explanation of the limitations.

---

> > > ### Author Response · Authors · 2026-04-05
> > >
> > > Q1: Thank you for the detailed response. I would like to see a more thorough explanation of the limitations.
> > >
> > > A1: Thank you for the follow-up. We agree that the limitations should be stated more explicitly, and we will incorporate a concise discussion into the revised paper.
> > >
> > > The current benchmark is designed for long-horizon, table-centric data analysis, with emphasis on table retrieval, preprocessing of structurally complex tables (e.g., decomposition and header merging), and programming-based data extraction. Consequently, its present scope is limited to table-only scenarios. This design does not yet cover spreadsheets containing visually grounded elements, such as embedded images, which would require dedicated vision tools and more specialized processing pipelines. In addition, our current table preview mechanism mainly relies on Markdown conversion, which is effective for text-dominant tables but less suitable for richer visual structures. More broadly, the benchmark currently focuses on analysis tasks and does not yet extend to spreadsheet-operation scenarios such as content filling, validation, deletion, formatting adjustment, or style editing. These limitations point to important directions for future extensions, particularly toward multimodal and operation-oriented table-agent settings. These directions also define an important avenue for our next stage of research.

---

### Official Review · Reviewer_MHVv · 2026-03-11

**Soundness:** 2
**Presentation:** 3
**Significance:** 2
**Originality:** 3
**Overall Recommendation:** 3
**Confidence:** 4

**Summary:**

This paper introduces TableAgent-Bench, a large-scale, bilingual multi-turn table QA benchmark tailored for industrial scenarios. The core innovation of this benchmark lies in its topology-aware dialogue construction strategy. An experimental evaluation of 25 mainstream LLMs reveals a significant capability gap in current models regarding complex industrial tabular reasoning.

**Compliance With Llm Reviewing Policy:**

Affirmed.

**Final Justification:**

I have read the authors’ response. I understand the authors’ point that allowing error propagation can reveal limitations of the model itself. However, I think it is very important to have clear mechanisms to prevent or recover from error propagation in multi-turn interactions. Given that this is a practical limitation of the current framework, I choose to keep my score.

**Key Questions For Authors:**

1. The ablation experiments in Table 5 show that the strongest model, Claude-Haiku-4-5, experienced a decrease in IC and Avg@3 after adding all tools. The authors speculate this is due to "functional redundancy" and "localization noise." Could you explain this phenomenon in more detail?
2. I would like to ask about the main situations in which the three models disagreed during the evaluation process, as the paper used three LLMs as evaluators for majority voting. Is there a mechanism to handle situations where a majority vote could not be achieved?
3. Figure 6 in the paper shows a high proportion of "propagation errors." Does the current TAEF framework have a mechanism to block error propagation in multi-round interactions?
4. I think that in real-world industrial applications, partially correct intermediate results or close values also have some value. Have the authors considered this aspect? I feel that the current benchmark is quite stringent.

**Limitations:**

yes

**Strengths And Weaknesses:**

*Strengths*

- Most existing table QA benchmarks are limited to single-turn queries or academic tables. This work constructs a benchmark specifically formulti-turn table QA within industrial contexts, which holds significant real-world value.
- The proposed "topology-aware" dialogue construction strategy systematically addresses complex cross-table dependencies and the evolution of user intent. The methodological design is well-reasoned and robust.

*Weaknesses*

- The primary contribution of this paper is the construction of a benchmark. While the topology-aware construction method is novel, the work as a whole falls strictly within the scope of a benchmark paper, and themethodological innovation is somewhat lacking.
- The response quality for sub-questions is determined viamajority votingfrom three LLM judges. Although this is supported by existing literature, LLM-based evaluation can introduce inherent biases and consistency issues. The paper fails to sufficiently analyze disagreements between the judge models or provide a detailed comparison with human judgment. It reports the Cohen's Kappa without an in-depth discussion of disagreement cases.
- The paper points out that the strongest model has an IC of only 53.4% ​​and an Avg@3 score of only 5%, which is a very low baseline. While Figure 6 and Appendix A summarize the four types of errors, the analysis of the deeper reasons why the model performs so poorly on these tasks is insufficient. I think the paper remains at the classification level and lacks a more fundamental exploration. Adding this section would significantly improve the paper's quality.

---

> ### Author Rebuttal · Authors · 2026-03-31
>
> Dear Reviewer MHVv,
>
> Thanks for your time to review this paper. We would like to address your concerns as follows:
>
> **W1: Lack of methodology Innovation.**
>
> **A1:** Thanks. We clarify that TableAgent-Bench is not merely a dataset, it provides a complete evaluation infrastructure for industrial-scale table agents, encompassing: (1) a topology-aware dialogue construction method, (2) a specialized toolset for messy industrial tables, (3) a multi-dimensional evaluation framework, and (4) evalution-driven insights that guide future method design. Although agent training algorithms are beyond this work’s scope, we believe this benchmark provides a rigorous foundation for the community.
>
> **W2&Q2: LLM judge bias of voting and need disagreement analysis.**
>
> **A2:** Thanks. LLM-based evaluation with majority voting is standard practice in LLM-as-Judge, and the Cohen’s Kappa of 0.851 indicates strong agreement with human judgment. All evaluations are grounded in human-reviewed structured scoring criteria.
>
> A typical disagreement arises when the model provides the correct core answer but misses a supporting detail. For example, for a question asking for the lowest-value procurement record, the model correctly identifies the record and reports the total amount (¥900) but omits the unit price (¥9). In such cases, some judges reward the correct conclusion, while others require full completeness.
>
> **W3: The metrics(IC, Avg@3) are low need more Root Cause Analysis.**
>
> **A3:** Thanks. The seemingly low baseline performance mainly reflects both model capability gaps (Table 4) and intrinsic task complexity. Table 4 shows clear differences between reasoning vs. non-reasoning models and agentic-enhanced vs. standard models, which are reflected in interaction turns, tool use, CR, and TRR/TRP.
>
> We also identify three major failure modes:
> 1. Complex Table Comprehension (Explain Understanding Errors): Despite providing production-grade preprocessing tools for complex table structures , agents struggle to adaptively select appropriate tools. This rigidity causes schema extraction errors that propagate to downstream tasks（e.g., data filtering, code generation）.
> 2. Ambiguous Query Localization (Explain Retrieval Errors): synthetic metrics, vague descriptions, and cross-lingual references often trigger retrieval loops (repeatedly searching the same directories) or retrieval hallucinations (assuming table existence without verification).
> 3. Myopic Debugging (Explain Propagation Errors): In long-horizon tasks, agents patch local execution errors fixing but ignore global constraints, leading to unrecoverable logical inconsistencies.
>
> **Q1：More explanation for  "functional redundancy" and "localization noise."**
>
> **A1:** Thanks. The performance drop on strong models (e.g., Claude) stems from a mismatch between native coding ability and specialized toolsets:
> 1. Functional Redundancy (Over-engineering): Strong models natively excel at table processing via Python. Providing specialized Table Processing (TP) tools creates "functional overlap," often leading agents to unnecessarily invoke tools on simple tables. This over-complication can inadvertently corrupt correct schemas and cause subsequent code failures.
> 2. Localization Noise (Context Dilution): Excessive TP-tool use creates too many intermediate artifacts, which dilute the main instruction and weaken global task orientation.
>
> **Q3：Does the current TAEF framework have a mechanism to block error propagation?**
>
> **A3:** Thanks. We deliberately omitted error-blocking mechanisms in the TAEF framework. Importantly, our work is a benchmark rather than a methodological innovation. The fundamental goal of TableAgent-Bench is to evaluate the intrinsic agentic capabilities of LLMs(e.g. specifically long-horizon reasoning, self-reflection, and state-correction) to provide some insights for future model training. To ensure evaluation fairness and validity, we kept the framework strictly minimal, avoiding complex framework-level error-correction engineering. Allowing errors to propagate naturally is essential to strip away external engineering tricks and accurately expose current models' true deficits in native error isolation.
>
> **Q4: correct intermediate results also have some value and the current benchmark is stringent.**
>
> **A4:** We agree that partially correct results hold practical value, which is exactly why we introduced Information Coverage (IC) to award partial credit (where Gemini-3-Pro achieves 53.4%). We deliberately retained the stringent Avg@3 metric as it aligns with the ultimate goal of agentic data analysis: full autonomy. Professionals using tools like Claude Code or OpenClaw expect flawless results; the prohibitive cost of manual verification would entirely negate the value of automation. Given the zero-tolerance for errors in high-stakes fields (e.g., finance, medicine), this strict metric is essential to expose the true gap between current LLMs and reliable industrial agents.

---

> > ### Author Rebuttal · Reviewer_MHVv · 2026-04-02
> >
> > I have read the authors’ response. I understand the authors’ point that allowing error propagation can reveal limitations of the model itself. However, I think it is very important to have clear mechanisms to prevent or recover from error propagation in multi-turn interactions. Given that this is a practical limitation of the current framework, I choose to keep my score.

---

> > > ### Author Response · Authors · 2026-04-05
> > >
> > > Thank you for the clarification. We respect the reviewer's perspective and agree that recovery mechanisms are important for deployment-oriented agent systems. We are actively investigating a more systematic approach to long-horizon reasoning for table agents in our ongoing follow-up work. However, TableAgent-Bench is intentionally designed as an evaluation benchmark rather than an error-correction framework, since adding framework-level recovery would partially mask the model's native long-horizon robustness. Our contribution lies in the evaluation infrastructure itself—topology-aware task construction, a realistic tool environment, fine-grained metrics, and diagnostic findings.

---

### Official Review · Reviewer_BtHa · 2026-03-12

**Soundness:** 3
**Presentation:** 2
**Significance:** 2
**Originality:** 3
**Overall Recommendation:** 4
**Confidence:** 3

**Summary:**

Existing benchmarks only test single-turn table question answering, treating it as a passive natural language understanding task, without considering the ability of autonomous reasoning and tool-call in real-world scenarios.
To bridge this gap, the authors proposed TableAgent-Bench. Through a topology-aware dialogue construction strategy, the agent generate sub-questions and multi-turn dialogues, achieving proactive agent interaction. In addition, the authors built the Table-centric Agent Evaluation Framework (TAEF), which includes a sandbox environment equipped with a toolset and four evaluation metrics. TAEF records the complete trajectory, allowing systematic diagnosis of intermediate failures and evaluation of performance in table localization, tool invocation, and trajectory-level pass rate. Finally, the authors tested 25 state-of-the-art models and found that even the best-performing models are still unreliable.

**Compliance With Llm Reviewing Policy:**

Affirmed.

**Key Questions For Authors:**

1) Why are the Avg@3 scores of all LLMs very low? Among the test results of the 25 state-of-the-art LLMs, taking Gemini-3-Pro-Preview as an example, the Completion Rate (CR) reached 100%, but Avg@3 was only 5%. Does this indicate that the subproblem is unreasonable?

**Limitations:**

The author did not mention the limitations in the work. I think there are the following two points:
1) In the Table-centric Agent Evaluation Framework (TAEF), sub-questions in multi-turn dialogue are directly input to the Table Agent. This leads to a lack of testing of the agent's planning ability, which is the foundation for performing subsequent tasks.
2) Among the test results of the 25 state-of-the-art LLMs, taking Gemini-3-Pro-Preview as an example, the Completion Rate (CR) reached 100%, but Avg@3 was only 5%. The author did not discuss why there is such a large gap between the two metrics.

**Strengths And Weaknesses:**

Strengths
High innovativeness: The article innovatively proposes constructing a multi-turn dialogue dataset and a testing framework. It also refines evaluation metrics down to the tool invocation level. It is inspirational for subsequent work.
Scientific and rational experiments: There is a high correlation between research challenges and experimental design. The experimental setup is reasonable, and authors further discuss ablation experiments, the effects of context length and interaction steps.
Reliable and rich data: there are many tables in the experimental dataset, covering a wide range of fields. Human involvement in reviewing multi-turn dialogue trajectories improves data reliability.
Weaknesses
Unreasonable task design: In the Table-centric Agent Evaluation Framework (TAEF), sub-questions in multi-turn dialogue are directly input to the Table Agent, instead of being generated by Table Agent. This leads to a lack of testing of the agent's planning ability, which is the foundation for performing subsequent tasks.
Insufficient experimental analysis: Among the test results of the 25 state-of-the-art LLMs, taking Gemini-3-Pro-Preview as an example, the Completion Rate (CR) reached 100%, but Avg@3 was only 5%. The author did not discuss why there is such a large gap between the two metrics.

---

> ### Author Rebuttal · Authors · 2026-03-31
>
> Dear Reviewer BtHa,
>
> Thank you for taking the time to review this paper and we appreciate the positive feedback and your detailed questions regarding evaluation framework and metrics. We would like to address your concerns as follows:
>
> **Q1: In the Table-centric Agent Evaluation Framework (TAEF), sub-questions in multi-turn dialogue are directly input to the Table Agent. This leads to a lack of testing of the agent's planning ability, which is the foundation for performing subsequent tasks.**
>
> **A1:** We thank the reviewer for this insightful observation. While macro-planning is crucial, directly inputting sub-questions was a deliberate design choice driven by two considerations:
>
> 1. Intra-Turn Complexity and Agentic Planning and Reasoning: Beyond inter-turn topological dependencies, each sub-question itself possesses sufficient analytical depth—encompassing cross-table queries, multi-intent combinations, and comprehensive comparative analyses, rather than simplistic lookups. Therefore, even within a single turn, agents must autonomously perform task planning and decomposition, demonstrating that predefined skeletons do not diminish the demand for advanced reasoning capabilities.
>
> 2. A Necessary Trade-off to Focus on Long-Horizon Tool Use: The primary objective of our benchmark is to rigorously evaluate an agent's long-horizon tool invocation capabilities (specifically table retrieval, processing, and analysis). While we initially considered fully dynamic multi-turn interactions, comprehensive data analysis is inherently open-ended. For instance, an abstract task like "Analyze German exported cars" has unbounded analytical angles (e.g., analyzing price trends, geographic distribution, or sales volume). In a dynamic setting, models would expend excessive effort on macro-level planning to explore these directions. This over-emphasis on macro-planning would dilute our primary focus on assessing actual tool-execution capabilities and make establishing a unique ground truth impossible. Consequently, we made a deliberate design trade-off by choosing topology-based static dialogues. This offloads unbounded macro-planning, ensuring a deterministic evaluation of the agent's core table-centric tool use while still preserving the structural complexity of human analytical thinking.
>
> **Q2: Among the test results of the 25 state-of-the-art LLMs, taking Gemini-3-Pro-Preview as an example, the Completion Rate (CR) reached 100%, but Avg@3 was only 5%. The author did not discuss why there is such a large gap between the two metrics.**
>
> **A2**: Thank you for this insightful observation. The gap exists because CR measures whether the agent finishes the task without crashing, while Avg@3 measures whether it gets *every score point correct* throughout the multi-turn dialogue. An agent can successfully complete all steps (high CR) but still produce wrong answers due to errors in table retrieval, data understanding, or code generation (low Avg@3). We elaborate on three key points:
>
> 1. CR Reflects Robustness, Not Accuracy:
> CR (Completion Rate) measures whether an agent completes all sub-questions without exceeding the 70-step limit. CR fails only when agents get stuck in: (1) Retrieval loops: repeated failures due to ambiguous metrics, cross-lingual retrieval, or domain terminology confusion. (2) Code debugging loops: continuous code execution failures. As Appendix B shows, agents spend nearly 84% of steps on table retrieval and coding. SOTA models like Gemini-3-Pro have strong self-debugging capabilities and efficient tools (semantic search, exact matching), so they rarely get trapped and achieve high CR. However, completing steps doesn't guarantee correct answers—agents can finish without crashing but still produce wrong results.
>
> 2. Avg@3 is Stringent by Design:
> Avg@3 is a trajectory-level metric, it requires *every* answer in the multi-turn dialogue to be correct. A single numerical error fails the entire trajectory. This strictness is intentional, i.e., in real-world data analysis, accuracy is non-negotiable. If users still need to verify results manually, the automation value is lost.
>
> 3. Why "Completed but Wrong" Happens (Section 5.3):
> In Section 5.3, we systematically identify four primary sources contributing to the "completed but wrong" phenomenon. Localization fails when ambiguous filenames or deep directories hinder table finding. Understanding errors arise from misinterpreting query requirements or domain indicators. Coding suffers from repeated failures due to complex table structures. Finally, error propagation causes early mistakes to cascade to downstream steps.
>
> In summary, TableAgent-Bench reveals that while LLM agents can execute long workflows reliably (high CR), they still struggle with accurate end-to-end data analysis (low Avg@3). This gap highlights the challenge of deploying agents in real-world industrial scenarios where accuracy is critical.

---

### Official Review · Reviewer_wBng · 2026-03-16

**Soundness:** 3
**Presentation:** 3
**Significance:** 3
**Originality:** 3
**Overall Recommendation:** 5
**Confidence:** 4

**Summary:**

This paper presents a new benchmark for (agentic) LLMs. The setup is that starting from a number of relevant tables, the LLMs have to answer a number of user questions in a sequential dialogue to reach a final answer. To do so they have access to some tools (retrieval, table processing, code execution) and must use these and their own reasoning to extract an answer.

The benchmark generation is done in 3 steps: first tables are collected from the web and filtered.

The second step is the most crucial: A dialogue is constructed by picking a task-template (like trend analysis) and a topology that defines an information flow graph where nodes are sub questions and the directed edges are dependencies. From that they use an LLM to generate multi-step QA trajectories, sampled and modified several times.

The (table set, task, QA sequences) tuples are again filtered for quality, both automatically and manually. Experts also produce a scoring schema for sub-question answers.

Evaluation is done not only on the final answer but by checking if tools were called successfully, quality of intermediate answer to the users's sub-questions via an LLM judge, checking if the correct tables from the set were used, and 'trajectory' which combines final answer quality and how hard the final answer was to reach.

Experiments are done across many current models, using shared prompt for the agentic functionalities. Results show that even the best models don't saturate the task yet. Interestingly, even among the best models, the number of tool calls varies by 2x.

**Compliance With Llm Reviewing Policy:**

Affirmed.

**Final Justification:**

I am satisfied with the rebuttal answers, checked the relevant paragraphs in the paper again and would recommend this work for publication, with some minor updates as discussed with the other reviewers.

**Key Questions For Authors:**

All my questions are around the selection and retrieval of table sets

[1] When collecting the tables - is there an effort to find sets of tables that belong together?

[2] When generating tasks, how is the set of relevant tables defined?

[3] In the benchmark, are only relevant tables given in a task? Does the agent initially receive a summary description of each table?

**Limitations:**

There is no limitations section in the paper and I am not sure one is needed.

**Strengths And Weaknesses:**

This well written and overall strong paper presents a complex benchmark for the current generations of agentic LLMs.

Much attention has been devoted to make this a realistic task that lies on the intersection of table-based and agentic benchmarks.

Evaluation relies not purely on the correctness of the final answer but also evaluates intermediate steps. There are some nuances to the experimental setup that could make the results even stronger. For example fixing the temperature at 0 might hurt some of the thinking models that expect a non-zero temperature. Also some LLMs have special tool calling capabilities, maybe these could be included - but I understand that tuning prompts for individual models might lead to apples to oranges comparisons.

Reproducibility of this work should be high and indeed the authors were able to test 25 LLMs with their benchmark.

---

> ### Author Rebuttal · Authors · 2026-03-31
>
> Dear Reviewer wBng,
>
> Thank you for taking the time to review this paper and we appreciate the positive feedback and your detailed questions regarding the dataset construction and evaluation methodology. We would like to address your concerns as follows:
>
> **Q1: When collecting the tables - is there an effort to find sets of tables that belong together?**
>
> **A1**:Yes. We strictly avoided artificially combining unrelated tables. Instead, our data collection process was inherently theme-driven. We preserved the natural physical and semantic boundaries of the data sources. As mentioned in Sections 3.1(line 144) and 3.4(line 243), our dataset covers 6 major domains and 28 subdomains. During the collection phase, tables were inherently grouped based on the same business theme or analytical scenario. Specifically, our multi-table sets are organized into cohesive folders that reflect real-world logical associations, with typical examples as follows:
>
> - Enterprise Finance: Multi-sheet workbooks belonging to a single report (e.g., a quarterly budget folder containing interrelated "Balance Sheet", "Sales Summary", and "Labor Costs").
> - Public Data: Records exported from the same statistical event (e.g., "College Entrance Examination Admission Scores").
> - Macroeconomics: Thematic reports under a unified topic (e.g., "2015-2019 Home Appliance Sales and Inventory Data"), and so on.
>
> By preserving these natural semantic and physical boundaries of tabular data, we ensure that the cross-table operations required in our benchmark fully reflect the genuine analytical logic of real-world data analysis tasks.
>
> **Q2: When generating tasks, how is the set of relevant tables defined?**
>
> **A2**: The relevant tables for each task are determined through a two-stage pipeline:
>
> In Stage 1 (Task Generation with Initial Table Assignment), building upon the scenario-based table organization discussed in Q1, we formulate the task input by combining all tables within a specific scenario folder with a randomly sampled task template (out of 10, e.g., comparative/trend analysis) and a QA structure (out of 6, e.g., Chain, Tree). We then provide the corresponding table summaries to the LLM. Guided by these sampled conditions, the LLM autonomously designs multi-turn dialogues and dynamically selects the relevant tables—either a subset or the entirety of the folder. Ultimately, this stage outputs the generated multi-turn tasks paired with their initially assigned, LLM-predicted relevant tables.
>
> In Stage 2 (Human Verification for Ground Truth), as detailed in Section 3.3 (Line 182), we establish the exact ground truth $G$ through a rigorous dual quality control strategy. This process begins with an automated validation phase, where the system preliminarily screens the associative relationships between the generated answers and candidate tables, effectively demarcating and narrowing the scope for human review. Building upon this preliminary filtering, professional reviewers then conduct expert annotation. They meticulously examine each multi-table collection and manually label the original source tables for every data component present in the generated answers, thereby finalizing the robust ground truth $G$.
>
> **Q3: In the benchmark, are only relevant tables given in a task? Does the agent initially receive a summary description of each table?**
>
> **A3**: In our benchmark, we do not restrict the input to only relevant tables, nor do we provide summary descriptions of the tables. For each task, the agent is only provided with the task query and the root file path of the table database. The agent must autonomously execute the entire workflow: browsing the file directory, locating relevant tables, understanding their content, and writing code to solve the problem.
>
> The core design philosophy of TableAgent-Bench is to evaluate the autonomous capabilities of LLM agents in real-world scenarios ("in the wild"). In actual work environments, human analysts facing data queries must similarly go through the complete process of searching for data across folders, reading raw tables, and deriving answers. Our benchmark specifically evaluates an agent's long-horizon problem-solving capabilities in these realistic settings, where trajectories average 30+ tool invocations and accumulate ~300K tokens in context. For example, when a user employs an agent framework (e.g., OpenDevin or Open Interpreter), they typically mount it to a root directory containing massive amounts of data and ask a question directly. The agent must independently use terminal commands (e.g., `ls`) to browse the directory structure, read the first few rows (e.g., `df.head()`) to understand the table schema, and finally write code to answer the query. Therefore, we deliberately excluded any manual preprocessing (such as filtering irrelevant tables or generating table summaries) to preserve the inherent complexity and realistic challenges of the tasks.

---

> > ### Author Rebuttal · Reviewer_wBng · 2026-04-02
> >
> > Thanks for extensive explanations.
> >
> > I have one more question regarding the table retrieval by the agent. From Table 3 (Toolset) I had guessed that tables were mostly selected by keyword of semantic retrieval. What is the role of file system operations here? Can an agent look for others files in the directory? Is there a notion of filesystem proximity?

---

> > > ### Author Response · Authors · 2026-04-05
> > >
> > > **Q1**: I have one more question regarding the table retrieval by the agent. From Table 3 (Toolset) I had guessed that tables were mostly selected by keyword of semantic retrieval. What is the role of file system operations here? Can an agent look for others files in the directory? Is there a notion of filesystem proximity?
> > >
> > > **A1**: File-system operations play a complementary but important role in TableAgent-Bench because the benchmark is designed to mimic how analysts work with raw table collections rather than pre-retrieved inputs. In realistic settings, related tables are often organized into small scenario-level folders, and our benchmark preserves this property. Therefore, agents can use command-line operations such as `ls` to inspect the directory hierarchy and explore neighboring files.
> > >
> > > However, directory browsing alone is insufficient because the repository can contain a highly complex directory structure, which can consume substantial agent context. For this reason, retrieval in our benchmark is not limited to semantic search. Instead, agents typically combine several concrete tools, including command-line directory browsing (`ls`), keyword-based exact matching (`grep`-style search), table-level semantic retrieval, and row-/column-level semantic retrieval. In practice, these four mechanisms are complementary:
> > >
> > > (1) Directory browsing. Agents first inspect folders and filenames to form an initial hypothesis about where relevant tables may reside. In our case analysis, they often begin with such exploration and may revisit the local directory of a partially relevant table when they need more precise evidence.
> > >
> > > (2) Keyword-based retrieval. Exact matching is useful when directory listings are too long to inspect efficiently, or when the question contains explicit indicators such as metric names, rankings, or identifiable entities. Its limitation is that it may return many candidates and is less robust for vague or underspecified indicators.
> > >
> > > (3) Table-level semantic retrieval. Semantic table selection helps when the directory structure is too large to explore exhaustively or when keyword retrieval repeatedly fails due to ambiguous wording.
> > >
> > > (4) Row- and column-level semantic retrieval. Once the agent has approximately localized the right table, row/column retrieval helps quickly identify the relevant records and fields. In our experience, this is often more effective than keyword matching for vague indicators because it operates closer to the table content and schema.
> > >
> > > Therefore, the agent can inspect other files in the directory, and filesystem proximity does exist. However, it serves only as a weak structural prior: nearby files are more likely to be relevant, but correctness is still evaluated against the human-verified relevant table set rather than folder co-location.

---

### Decision · Program_Chairs · 2026-04-30

**Decision:**

Accept (regular)

**Comment:**

This paper introduces TableAgent-Bench, a benchmark for evaluating LLM agents on multi-turn table question answering tasks involving complex table collections and tool-based reasoning. The benchmark constructs multi-step dialogues grounded in industrial-scale tables and provides a dedicated evaluation framework assessing intermediate tool use, table localization, and trajectory-level correctness. Experiments across 25 state-of-the-art models reveal substantial gaps in current agents’ ability to perform long-horizon table reasoning.

Strengths: Reviewers agree that the paper addresses an important gap in existing TableQA benchmarks, which typically focus on single-turn questions rather than agentic multi-step reasoning. The dataset construction pipeline, topology-aware dialogue design, and tool-based evaluation framework provide a realistic evaluation setting for table-centric agents. The large scale of the dataset, the inclusion of industrial-style tables, and the evaluation across many models contribute to the paper’s potential value as a benchmark resource for the community.

Weaknesses and Remaining Concerns:  Some reviewers noted that the work is primarily a benchmark contribution, with limited methodological novelty beyond dataset and evaluation design. Questions were also raised regarding aspects of the evaluation framework, including reliance on LLM-based judges, limited analysis of disagreements between judges, and relatively shallow analysis of the causes of model failures. Another concern is that sub-questions are provided to the agent rather than generated by the agent, which may limit evaluation of planning capabilities. While the rebuttal clarified several design choices and evaluation details, one reviewer remained unconvinced on some of these points.

Overall, reviewers generally agree that the benchmark addresses a relevant and under-explored evaluation setting for agentic table reasoning and provides a useful dataset and evaluation framework. Although concerns remain regarding the depth of analysis and methodological novelty, I think the strengths outweigh the weaknesses and the work could be potentially valuable for future research. The authors should carefully revise the paper by incorporating the discussions during the rebuttal.